# Learning Prototype-oriented Set Representations for Meta-Learning

**Dandan Guo [1], Long Tian[2], Minghe Zhang[3], Mingyuan Zhou[4], Hongyuan Zha[1]**
[1]The Chinese University of Hong Kong, Shenzhen   [2]Xidian University
[3]Georgia Institute of Technology    [4]The University of Texas at Austin
guodandan@cuhk.edu.cn tianlong@xidian.edu.cn mzhang388@gatech.edu
migyuan.zhou@mccombs.utexas.edu   zhahy@cuhk.edu.cn

## Abstract

Learning from set-structured data is a fundamental problem that has recently attracted increasing attention, where a series of summary networks are introduced to deal with the set input. In fact, many meta-learning problems can be treated as set-input tasks. Most existing summary networks aim to design different architectures for the input set in order to enforce permutation invariance. However, scant attention has been paid to the common cases where different sets in a meta-distribution are closely related and share certain statistical properties. Viewing each set as a distribution over a set of global prototypes, this paper provides a novel prototype-oriented optimal transport (POT) framework to improve existing summary networks. To learn the distribution over the global prototypes, we minimize its regularized optimal transport distance to the set empirical distribution over data points, providing a natural unsupervised way to improve the summary network. Since our plug-and-play framework can be applied to many meta-learning problems, we further instantiate it to the cases of few-shot classification and implicit meta generative modeling. Extensive experiments demonstrate that our framework significantly improves the existing summary networks on learning more powerful summary statistics from sets and can be successfully integrated into metric-based few-shot classification and generative modeling applications, providing a promising tool for addressing set-input and meta-learning problems.

## 1 Introduction

Machine learning models, such as convolutional neural networks for images (He et al. 2016) and recurrent neural networks for sequential data (Sutskever et al. 2014), have achieved great success in taking advantage of the structure in the input space (Maron et al. 2020). However, extending them to handle unstructured input in the form of sets, where a set can be defined as an unordered collections of elements, is not trivial and has recently attracted increasing attention (Jurewicz & Strømberg-Derczynski, 2021). Set-input is relevant to a range of problems, such as understanding a scene formed of a set of objects (Eslami et al. 2016), classifying an object composed of a set of 3D points (Qi et al. 2017), summarizing a document consisting of a set of words (Blei et al., 2003; Zhou et al., 2016), and estimating summary statistics from a set of data points for implicit generative models (Chen et al. 2021). Moreover, many meta-learning problems, which process different but related tasks, may also be viewed as set-input tasks (Lee et al., 2019), where an input set corresponds to the training dataset of a single task. Therefore, we broaden the scope of set-related applications by including traditional set-structured input problems and most meta-learning problems. Both of them aim to improve the quick adaptation ability for unseen sets, even though the latter is more difficult because of limited samples or the occurrence of new categories for classification problems.

For a set-input, the output of the model must not change if the elements of the input set are reordered, which entails permutation invariance of the model. To enforce this property, multiple researchers have recently focused on designing different network architectures, which can be referred to as a *summary network* for compressing the set-structured data into a fixed-size output. For example, the prominent works of Zaheer et al. (2017) and Edwards & Storkey (2017) combined the standard

feed-forward neural networks with a set-pooling layer, which have been proven to be universal approximators of continuous permutation invariant functions. Lee et al. (2019) further introduced Set Transformer to encode and aggregate the features within the set using multi-head attention. Maron et al. (2020) designed deep models and presented a principled approach to learn sets of symmetric elements. Despite the effectiveness and recent popularity of these works in set-input problems, there are several shortcomings for existing summary networks, which could hinder their applicability and further extensions: 1) The parameters of the summary network are typically optimized by a task-specific loss function, which could limit the models' flexibility. 2) A desideratum of a summary network is to extract set features, which have enough ability to represent the summary statistics of the input set and thus benefit the corresponding set-specific task; but for many existing summary networks, there is no clear evidence or constraint that the outputs of the summary network could describe the set's summary statistics well. These limits still remain even with the recent more carefully designed summary networks, while sets with limited samples further exacerbate the problem.

To address the above shortcomings, we present a novel and generic approach to improve the summary networks for set-structured data and adapt them to meta-learning problems. Motivated by meta-learning that aims to extract transferable patterns useful for all related tasks, we assume that there are $K$ global prototypes (*i.e.*, centers) among the collection of related sets, and each prototype or center is encouraged to capture the statistical information shared by those sets, similar to the "topic" in topic modeling (Blei et al., 2003; Zhou et al., 2016) or "dictionary atom" in dictionary learning (Aharon et al., 2006; Zhou et al., 2009). Specifically, for the $j$th set, we consider it as a discrete distribution $P_j$ over all the samples within the set (in data or feature space). At the same time, we also represent this set with another distribution $Q_j$ (in the same space with $P_j$), supported on $K$ global prototypes with a $K$-dimensional set representation $\boldsymbol{h}_j$. Since $\boldsymbol{h}_j$ measures the importance of global prototypes for set $j$, it can be treated as the prototype proportion for summarizing the salient characteristics of set $j$. Moreover, the existing summary networks can be adopted to encode set $j$ as $\boldsymbol{h}_j$ for their desired property of permutation invariance. In this way, we can formulate the learning of summary networks as the process of learning a $P_j$ to be as close to $Q_j$ as possible, a process facilitated by leveraging the optimal transport (OT) distance (Peyré & Cuturi 2019). Therefore, the global prototypes and summary network can be learned by jointly optimizing the task-specific loss and OT distance between $P_j$ and $Q_j$ in an *end-to-end* manner. We can refer to this method as prototype-oriented OT (POT) framework for meta-learning, which is applicable to a range of unsupervised and supervised tasks, such as set-input problems solved by summary networks, meta generation (Hong et al., 2020c; Antoniou et al., 2017), metric-based few-shot classification (Snell et al., 2017), and learning statistics for approximate Bayesian computation (Chen et al., 2021). We note our construction has drawn inspirations from previous works that utilize a transport based loss between a set of objects and a set of prototypes (Tanwisuth et al., 2021; Wang et al., 2022). These works mainly follow the bidirectional conditional transport framework of Zheng & Zhou (2021), instead of the undirectional OT framework, and focus on different applications.

Since our plug-and-play framework can be applied to many meta-learning problems, this paper further instantiates it to the cases of metric-based few-shot classification and implicit meta generative modeling. We summarize our contributions as follows: (1) We formulate the learning of summary network as the distribution approximation problem by minimizing the distance between the distribution over data points and another one over global prototypes. (2) We leverage the POT to measure the difference between the distributions for use in a joint learning algorithm. (3) We apply our method to metric-based few-shot classification and construct implicit meta generative models, where a summary network is used to extract the summary statistics from set and optimized by the POT loss. Experiments on several meta-learning tasks demonstrate that introducing the POT loss into existing summary networks can extract more effective set representations for the corresponding tasks, which can also be integrated into existing few-shot classification and GAN frameworks, producing a new way to learn the set' summary statistics applicable to many applications.

## 2 BACKGROUND

### 2.1 SUMMARY NETWORKS FOR SET-STRUCTURED INPUT

To deal with the set-structured input $D_j = \{\boldsymbol{x}_{j,1:N_j}\}$ and satisfy the permutation invariance in set, a remarkably simple but effective summary network is to perform pooling over embedding vectors extracted from the elements of a set. More formally,

$$S_\phi(D_j) = g_{\phi_2}\left(\text{pool}\left(\left\{f_{\phi_1}(\boldsymbol{x}_{j1}), \ldots, f_{\phi_1}(\boldsymbol{x}_{jN_j})\right\}\right)\right), \tag{1}$$

where $f_{\phi_1}(\cdot)$ acts on each element of a set and $g_{\phi_2}(\text{pool}(\cdot))$ aggregates these encoded features and produces desired output, and $\phi = \{\phi_1, \phi_2\}$ denotes the parameters of the summary network. Most network architectures for set-structured data follow this structure; see more details from previous works (Lee et al., 2019; Zaheer et al., 2017; Edwards & Storkey, 2017; Maron et al., 2020).

## 2.2 OPTIMAL TRANSPORT

Although OT has a rich theory, we limit our discussion to OT for discrete distributions and refer the reader to Peyré & Cuturi (2019) for more details. Let us consider $p$ and $q$ as two discrete probability distributions on the arbitrary space $X \subseteq \mathbb{R}^d$, which can be formulated as $p = \sum_{i=1}^{n} a_i \delta_{x_i}$ and $q = \sum_{j=1}^{m} b_j \delta_{y_j}$. In this case, $\boldsymbol{a} \in \Sigma^n$ and $\boldsymbol{b} \in \Sigma^m$, where $\Sigma^n$ denotes the probability simplex of $\mathbb{R}^n$. The OT distance between $\boldsymbol{a}$ and $\boldsymbol{b}$ is defined as

$$\text{OT}(\boldsymbol{a}, \boldsymbol{b}) = \min_{\mathbf{T} \in U(\boldsymbol{a}, \boldsymbol{b})} \langle \mathbf{T}, \mathbf{C} \rangle, \tag{2}$$

where $\langle \cdot, \cdot \rangle$ means the Frobenius dot-product; $\mathbf{C} \in \mathbb{R}_{\geq 0}^{n \times m}$ is the transport cost function with element $C_{ij} = C(x_i, y_j)$; $\mathbf{T} \in \mathbb{R}_{>0}^{n \times m}$ denotes the doubly stochastic transport probability matrix such that $U(\boldsymbol{a}, \boldsymbol{b}) := \{\mathbf{T} \mid \sum_{i}^{n} T_{ij} = b_j, \sum_{j}^{m} T_{ij} = a_i\}$. To relax the time-consuming problem when optimising the OT distance, Cuturi (2013) introduced the entropic regularization, $H = -\sum_{ij} T_{ij} \ln T_{ij}$, leading to the widely-used Sinkhorn algorithm for discrete OT problems.

## 3 PROPOSED FRAMEWORK

In meta-learning, given a meta-distribution $p_{\mathcal{M}}$ of tasks, the marginal distribution $p_j$ of task $j$ is sampled from $p_{\mathcal{M}}$ for $j \in \mathcal{J}$, where $\mathcal{J}$ denotes a finite set of indices. E.g., we can sample $p_j$ from $p_{\mathcal{M}}$ with probability $\frac{1}{\mathcal{J}}$ when $p_{\mathcal{M}}$ is uniform over a finite number of marginals. During meta-training, direct access to the distribution of interest $p_j$ is usually not available. Instead, we will observe a set of data points $D_j = \{\boldsymbol{x}_{ji}\}_{i=1}^{N_j}$, which consists of $N_j$ i.i.d. samples from $p_j$ over $\mathbb{R}^d$. We can roughly treat the meta-learning problems as the set-input tasks, where dataset $D_j$ from $p_j$ corresponds to an input set. To learn more representative features from related but unseen sets in meta-learning problems, we adopt the summary network as the encoder to extract set representations and improve it by introducing the OT loss and global prototypes, providing many applications. Besides, we also provide the applications to metric-based few-shot classification and implicit generative framework by assimilating the summary statistics. Below we describe our model in detail.

### 3.1 LEARNING GLOBAL PROTOTYPES AND SET REPRESENTATION VIA OT

Given $J$ sets from meta-distribution $p_{\mathcal{M}}$, we can represent each set $D_j$ from meta-distribution $p_{\mathcal{M}}$ as an empirical distribution over $N_j$ samples on the original data space, formulated as

$$P_j = \sum_{i=1}^{N_j} \frac{1}{N_j} \delta_{\boldsymbol{x}_{ji}}, \boldsymbol{x}_{ji} \in \mathbb{R}^d. \tag{3}$$

Since all sets (distributions) drawn from meta-distribution $p_{\mathcal{M}}$ are closely related, it is reasonable to assume that these sets share some statistical information. Motivated by dictionary learning, topic modeling, and two recent prototype-oriented algorithms (Tanwisuth et al., 2021; Wang et al., 2022), we define the shared information as the learnable global prototype matrix $\mathbf{B} = \{\boldsymbol{\beta}_k\} \in \mathbb{R}^{d \times K}$, where $K$ represents the number of global prototypes and $\boldsymbol{\beta}_k$ denotes the distributed representation of the $k$-th prototype in the same space of the observed data points (e.g., "topic" in topic modeling). Given the prototype matrix $\mathbf{B}$, each set can be represented with a $K$-dimensional weight vector $\boldsymbol{h}_j \in \Sigma_k$ (e.g., "topic proportion" in topic modeling), where $h_{jk}$ means the weight of the prototype $\boldsymbol{\beta}_k$ for set $j$. Hence, we can represent set $D_j$ with another distribution $Q_j$ on prototypes $\boldsymbol{\beta}_{1:K}$:

$$Q_j = \sum_{k=1}^{K} h_{jk} \delta_{\boldsymbol{\beta}_k}, \boldsymbol{\beta}_k \in \mathbb{R}^d, \tag{4}$$

where $\boldsymbol{h}_j$ is a set representation for describing set $j$. Since set $j$ can be represented as $Q_j$ and $P_j$, we can learn set-specific representation $\boldsymbol{h}_j$ and prototype matrix $\mathbf{B}$ by pushing $Q_j$ towards $P_j$:

$$\text{OT}(P_j, Q_j) = \min_{\mathbf{B}, \boldsymbol{h}_j} \langle \mathbf{T}, \mathbf{C} \rangle \overset{\text{def.}}{=} \sum_{i}^{N_j} \sum_{k}^{K} C_{ik} T_{ik}, \tag{5}$$

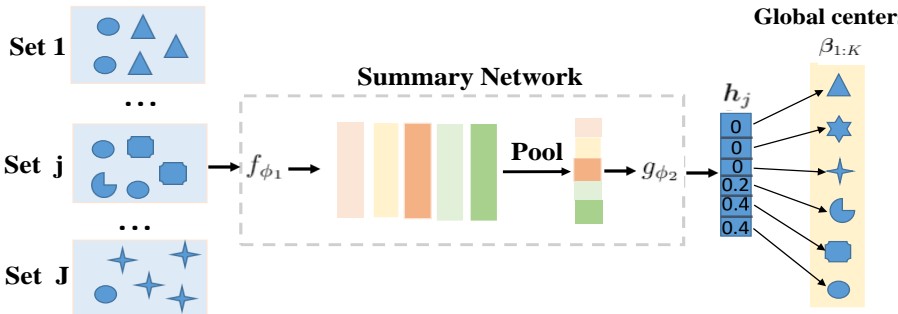

Figure 1: An overview of our proposed framework, where "pool" operation including mean, sum, max or similar. We compute the representation $\boldsymbol{h}_j$ for set $j$ using the summary network in Equation 1, which is the weight vector of the global prototypes (*i.e.*, centers) $\boldsymbol{\beta}_{1:K}$ in the corresponding set.

where $\mathbf{C} \in \mathbb{R}_{\geq 0}^{N_j \times K}$ is the transport cost matrix. In this paper, to measure the distance between data point $\boldsymbol{x}_{ji}$ in set $j$ and prototype $\boldsymbol{\beta}_k$, unless specified otherwise, we construct $\mathbf{C}$ as $C_{ik} = 1 - \cos(\boldsymbol{x}_{ji}, \boldsymbol{\beta}_k)$, which provides an upper-bounded positive similarity metric. Besides, the transport probability matrix $\mathbf{T} \in \mathbb{R}_{>0}^{N_j \times K}$ should satisfy $\Pi(\boldsymbol{a}, \boldsymbol{b}) := \left\{ \mathbf{T} \mid \mathbf{T}\mathbf{1}_K = \boldsymbol{a}, \mathbf{T}^\top \mathbf{1}_{N_j} = \boldsymbol{b} \right\}$ with $T_{ik} = T(\boldsymbol{x}_{ji}, \boldsymbol{\beta}_k)$, where $\boldsymbol{a} = [\frac{1}{N_j}] \in \Sigma^{N_j}$ and $\boldsymbol{b} = [h_{jk}] \in \Sigma^K$ denote the respective probability vectors for distribution $P_j$ in Equation 3 and $Q_j$ in Equation 4.

Since $\boldsymbol{h}_j$ should be invariant to permutations of the samples in set $j$, we adopt a summary network to encode the set of $N_j$ points. For unsupervised tasks, taking the summary network in Equation 1 as the example, we can directly add a Softmax activation function into $S_\phi$ to enforce the simplex constraint in set representation $\boldsymbol{h}_j$, denoted as $\boldsymbol{h}_j = \mathrm{Softmax}(S_\phi(D_j))$. As shown in Fig. 1, given $J$ sets, to learn the global prototype matrix $\mathbf{B}$ and summary network parameterized by $\phi$, we adopt the entropic constraint (Cuturi, 2013) and define the average OT loss for all training sets as

$$L_{\mathrm{POT}} = \min_{\mathbf{B}, \phi} \frac{1}{\mathcal{J}} \sum_{j=1}^{\mathcal{J}} \left( \sum_{i}^{N_j} \sum_{k}^{K} C_{ik} T_{ik} - \epsilon \sum_{i}^{N_j} \sum_{k}^{K} -T_{ik} \ln T_{ik} \right) = \min_{\mathbf{B}, \phi} \frac{1}{\mathcal{J}} \sum_{j=1}^{\mathcal{J}} \left( \mathrm{OT}_\epsilon(P_j, Q_j) \right), \quad (6)$$

where $\epsilon$ is a hyper-parameter for entropic constraint. Algorithm 1 describes the workflow of the POT loss for improving summary network under unsupervised tasks. For supervised tasks, set $j$ is denoted as $D_j = \{\boldsymbol{x}_{j,1:N_j}, \boldsymbol{y}_j\}$, where $\boldsymbol{y}_j$ is the ground-truth output determined by specific tasks. As $\boldsymbol{h}_j$ is a normalized weight vector, directly using it to realize the corresponding task may be undesired. Denoting $\boldsymbol{z}_j = \mathrm{pool}\left( \{f_{\phi_1}(\boldsymbol{x}_{j1}), \ldots, f_{\phi_1}(\boldsymbol{x}_{jN_j})\} \right)$, we project it to the following vectors:

$$\boldsymbol{h}_j = f_e(\boldsymbol{z}_j), \quad \hat{\boldsymbol{y}}_j = f_\lambda(\boldsymbol{z}_j), \quad (7)$$

where $\boldsymbol{h}_j$ and $\hat{\boldsymbol{y}}_j$ are responsible for the POT and task-specific losses, respectively. Now the summary network parameters $\tilde{\phi} = \{e, \lambda, \phi_1\}$ and global prototypes $\mathbf{B}$ are learned by jointly optimizing the task-specific loss (computed by $\hat{\boldsymbol{y}}_j$ and $\boldsymbol{y}_j$) and OT loss in Equation 6. In summary, minimizing the POT loss defined by the prototype distribution $Q_j$ and empirical distribution $P_j$ provides a principled and unsupervised way to encourage the summary network to capture the set's summary statistics. Therefore, our plug-and-play framework can integrate a suite of summary networks and realize efficient learning from new sets for both unsupervised and supervised tasks.

### 3.2 APPLICATION TO METRIC-BASED FEW-SHOT CLASSIFICATION

As a challenging meta-learning problem, few-shot classification has recently attracted increasing attention, where one representative method is metric-based few-shot classification algorithms. Taking the ProtoNet (Snell et al., 2017) as an example, we provide a simple but effective method to improve its classification performance with the help of POT and summary network, where we refer the reader to ProtoNet for more details. ProtoNet represents each class by computing an $M$-dimensional representation $\boldsymbol{z}_j \in \mathbb{R}^M$, with an embedding function $f_{\phi_1} : \mathbb{R}^d \to \mathbb{R}^M$, where we adopt the same $\phi_1$ as the learnable parameters, following the feature extractor in summary network in Equation 1 for simplicity. Formally, ProtoNet adopts the average pooling to aggregate the embedded features of the support points belonging to its class into vector $\boldsymbol{z}_j = \frac{1}{|S_j|} \sum_{(\mathbf{x}_{ji}, y_{ji}) \in S_j} f_{\phi_1}(\mathbf{x}_{ji})$. ProtoNet

---

**Algorithm 1** The workflow of POT on minimizing the OT distance between $P_j$ and $Q_j$.

---

**Require:** Datasets $\mathcal{D}_{1:J}$, batch size $m$, learning rate $\alpha$, initial summary network parameters $\phi$, initial global prototype matrix $\mathbf{B}$, cost function $C$ and hyper-parameter $\epsilon$.

**while** $\mathbf{B}, \phi$ has not converged **do**

    Randomly choose $j$ from $1, 2, .., \mathcal{J}$

    Sample the real data $\{\boldsymbol{x}_{ji}\}_{i=1}^m$ from set $j$, which is denoted as empirical distribution $P_j$ in Equation 3;

    Compute the set representation $\boldsymbol{h}_j = \mathrm{Softmax}(S_\phi(\{\boldsymbol{x}_{ji}\}_{i=1}^m))$ with summary network in Equation 1;

    Represent the $Q_j$ with global prototype matrix $\mathbf{B}$ and statistics $\boldsymbol{h}_j$ in Equation 4;

    Compute the loss $\mathrm{OT}_\epsilon(P_j, Q_j)$ between $P_j$ and $Q_j$ with Sinkhorn algorithm in Equation 6;

    $\mathbf{B} \leftarrow \mathbf{B} + \alpha g_{\mathbf{B}}$, where $g_{\mathbf{B}} \leftarrow \nabla_{\mathbf{B}} [\mathrm{OT}_\epsilon(P_j, Q_j)]$; $\phi \leftarrow \phi + \alpha g_\phi$, where $g_\phi \leftarrow \nabla_\phi [\mathrm{OT}_\epsilon(P_j, Q_j)]$;

**end while**

---

then compares the distance between a query point $f_{\phi_1}(\boldsymbol{x})$ to the $\mathbf{z}_j$ in the same embedding space. Motivated by the summary network, we further introduce a feed-forward network $g_{\phi_2}$ to map the $\mathbf{z}_j$ into the $\boldsymbol{h}_j$ used to define $Q_j$ distribution over $\mathbf{B}$. Therefore, the functions $g_{\phi_2}$ and $f_{\phi_1}$ can be jointly optimized by minimizing the POT loss and the original classification loss in ProtoNet,

$$\min_{\phi_1, \phi_2, \mathbf{B}} V(\phi_1, \phi_2, \mathbf{B}) = L_{\mathrm{POT}} + \sum_{j=1}^{\mathcal{J}} \sum_{i=1}^{N_j} \mathrm{CLS}(y_{ji}, \hat{y}_{ji}) \tag{8}$$

where $\boldsymbol{h}_j$ is used for computing the POT loss, $N_j$ the number of samples in class $j$, $\hat{y}_{ji}$ the predicted label for sample $\boldsymbol{x}_{ji}$, conditioned on $\mathbf{z}_{1:J}$ and $f_{\phi_1}(\boldsymbol{x}_{ji})$, and CLS the classification loss. Only introducing matrix $\mathbf{B}$ and $g_{\phi_2}$, whose parameters are usually negligible compared to $f_{\phi_1}$, our proposed method can benefit the metric-based few-shot classification by enforcing the $f_{\phi_1}$ to learn more powerful representation $\mathbf{z}_j$ of each class and feature $f_{\phi_1}(\boldsymbol{x})$ of the query sample.

## 3.3 APPLICATION TO IMPLICIT META GENERATIVE MODELS

Considering implicit meta generative modeling is still a challenging but important task in meta-learning, we further present how to construct the model by introducing set representation $\boldsymbol{h}_j$ as summary statistics, where we consider GAN-based implicit models. Specifically, given $p_j \sim p_{\mathcal{M}}$, we aim to construct a parametrized pushforward (*i.e.*, generator) of reference Gaussian distribution $\rho$, denoted as $T_\theta(\cdot, p_j)\sharp\rho$, to approximate the marginal distribution $p_j$, where $\theta$ summarizes the parameters of pushforward. Since it is unaccessible to the distribution of interest $p_j$, we replace $p_j$ with $P_j$ and use the summary network to encode set $D_j$ into $\boldsymbol{h}_j$ as discussed above, which is further fed into the generator serving as the conditional information, denoted as $T_\theta(\boldsymbol{z}; \boldsymbol{h}_j), \boldsymbol{z} \sim \rho$. To enforce the pushforward $T_\theta(\boldsymbol{z}; \boldsymbol{h}_j)$ to fit the real distribution $P_j$ as well as possible, we introduce a discriminator $f_w$ following the standard GAN (Goodfellow et al., 2014). Generally, our GAN-based model consists of three components. Summary network $S_\phi(\cdot)$ focuses on learning the summary statistics $\boldsymbol{h}_j$ by minimizing $\mathrm{OT}_\epsilon(P_j, Q_j)$. The pushforward aims to push the combination of a random noise vector $\boldsymbol{z}$ and statistics $\boldsymbol{h}_j$ to generate samples that resemble the ones from $P_j$, where we simply adopt a concatenation for $\boldsymbol{h}_j$ and $\boldsymbol{z}$ although other choices are also available. Besides, $f_w$ tries to distinguish the "fake" samples from the "real" samples in set $D_j$. Therefore, we optimize the implicit meta generative model by defining the objective function as:

$$\min_{\mathbf{B}, \phi, \theta} \max_w V(\mathbf{B}, \phi, \theta, w) = L_{\mathrm{POT}} + \frac{1}{\mathcal{J}} \sum_{j=1}^{\mathcal{J}} \mathbb{E}_{\boldsymbol{x} \sim P_j}[\log f_w(\boldsymbol{x})] + \mathbb{E}_{\boldsymbol{z} \sim \rho}[\log(1 - f_w(T_\theta(\boldsymbol{z}; \boldsymbol{h}_j)))] \tag{9}$$

In addition to the standard GAN loss, we can also adopt the Wasserstein GAN (WGAN) of Arjovsky et al. (2017) to approximate $P_j$. It is also flexibly to decide the input fed into $f_w$. For example, following the conditional GAN (CGAN) of Mirza & Osindero (2014), we can combine $\boldsymbol{h}_j$ and data points (generated or real), where the critic can be denoted as $f_w(\boldsymbol{x}, \boldsymbol{h}_j)$ and $f_w(T_\theta(\boldsymbol{z}; \boldsymbol{h}_j), \boldsymbol{h}_j)$, respectively. Since we focus on fitting the meta-distribution with the help of the summary network and POT loss, we leave the problem-specific design of the generator, critic, and summary network as future work for considerable flexibility in architectures. In Appendix A, we provide the illustration of our proposed model in Fig. 3 and detailed algorithm in Algorithm 2.

## 4 RELATED WORK

**Learning Summary Representation of Set-input.** There are two lines for learning the set representation. The first line aims to design more powerful summary networks, which are reviewed in

Introduction and Section 2.1 and omitted here due to the limited space. The another line assumes a/some to-be-learned reference set(s), and optimizes the distance between the original sets (or features of the observed data points) and the reference set(s) with OT or other distance measures, to learn set representation. For example, RepSet (Skianis et al., 2020) computes some comparison costs between the input sets and some to-be-learned reference sets with a network flow algorithm, such as bipartite matching. These costs are then used as set representation in a subsequent neural network. However, unlike our framework, RepSet does not allow unsupervised learning and mainly focuses on classification tasks. The Optimal Transport Kernel Embedding (OTKE) (Mialon et al., 2021) marries ideas from OT and kernel methods (Schölkopf et al., 2002), and aligns features of a given set to a trainable reference distribution. Wasserstein Embedding for Graph Learning (WEGL) (Kolouri et al., 2021) also uses a similar idea to the linear Wasserstein embedding as a pooling operator for learning from sets of features. To the best of our knowledge, both of them view the reference distribution as the barycenter and compute the set-specific representation by aggregating the features (embedded with kernel methods) in a given set with adaptive weight, defined by the transport plan between the given set and the reference. Different from them, we assume $J$ probability distributions (rather than one reference distribution with an uniform measure) over these shared prototypes by taking set representations $h_{1:J}$ as the measures, to approximate the corresponding $J$ empirical distributions, respectively. Then we naturally use the summary network as the encoder to compute $h_{1:J}$, which can be jointly optimized with the shared prototypes by minimizing the POT loss in an unsupervised way. For a given set, we can directly compute its representation with summary network, avoiding iteratively optimizing the transport plan between the given set and learned reference like OTKE and WEGL. These differences between the barycenter problem and ours, which are further described in Appendix B, lead to different views of set representation learning and different frameworks as well. To learn compact representations for sequential data, Cherian & Aeron (2020) blend contrastive learning, adversarial learning, OT, and Riemannian geometry into one framework. However, our work directly minimises the POT cost between empirical distribution and the to-be-learned distribution, providing a laconic but effective way to learn set representation.

**Metric-based few-shot classification methods.** Our method has a close connection with metric-based few-shot classification algorithms. For example, MatchingNet (Vinyals et al., 2016) and ProtoNet (Snell et al., 2017) learned to classify samples by computing distances to representatives of each class. Using an attention mechanism over a learned embedding of the support set to predict classes for the query set, MatchingNet (Vinyals et al., 2016) can be viewed as a weighted nearest-neighbor classifier applied within an embedding space. ProtoNet (Snell et al., 2017) takes a class's prototype to be the mean of its support set in the learned embedding space, which further performs classification for an embedded query point by finding the nearest class prototype. Importantly, the global prototypes in our paper are shared among all sets, which is different from the specific prototype for each class in ProtoNet but suitable for our case. Due to the flexibility of our method, we can project the average aggregated feature vector derived from the encoder in ProtoNet or MatchingNet, into $h_j$ by introducing a simple neural network, which can be jointly optimized with the encoder by minimizing the classification loss and POT loss. Our novelty is that the POT loss can be naturally used to improve the learning of encoder while largely maintaining existing model architectures or algorithms. Another recent work for learning multiple centers is infinite mixture prototypes (IMP) (Allen et al., 2019), which represents each class by a set of clusters and infers the number of clusters with Bayesian nonparametrics. However, in our work, the centers are shared for all classes and set-specific feature extracted from the summary network serves as the proportion of centers, where the centers and summary network can be jointly learned with the POT loss.

**Meta GAN-based Models.** As discussed by Hong et al. (2020a), meta GAN-based models can be roughly divided into optimization-based, fusion-based, and transformation-based methods. Clouâtre & Demers (2019) and Liang et al. (2020) integrated GANs with meta-learning algorithms to realize the optimization-based methods, including model-agnostic meta-learning (MAML) (Finn et al., 2017) and Reptile (Nichol et al., 2018). Hong et al. (2020b;c) fused multiple conditional images by combining matching procedure with GANs, providing fusion-based methods. For transformation based methods, Antoniou et al. (2017) and Hong et al. (2020a) combine only one image and the random noise into the generator to produce a slightly different image from the same category, without using the multiple images from same category. Besides, a recent work that connects existing summary network with GAN is MetaGAN (Zhang et al. 2018), which feeds the output of the summary network into the generator and focuses on few-shot classification using MAML. The key differences

of these models from ours is that we develop POT to capture each sets' summary statistics, where we can flexibly choose the summary network, generator, and discriminator for specific tasks.

## 5 EXPERIMENTS

We conduct extensive experiments to evaluate the performance of our proposed POT in improving summary networks, few-shot generation, and few-shot classification. Unless specified otherwise, we set the weight of entropic constraint as $\epsilon = 0.1$, the maximum iteration number in Sinkhorn algorithm as 200, and adopt the Adam optimizer (Kingma & Ba, 2015) with learning rate 0.001. We repeat all experiments 5 times and report the mean and standard deviation on corresponding test datasets.

### 5.1 EXPERIMENTS ABOUT POT LOSS IN SUMMARY NETWORK

To evaluate the effectiveness of POT in improving the summary network, we conduct three tasks on two classical architectures: DeepSets (Zaheer et al. 2017) and Set Transformer (Lee et al. 2019), where the former uses standard feed-forward neural networks and the latter adopts the attention-based network architecture. For Set Transformer and DeepSets, the summary network is defined in Equation 1 and optimized by the task-specific loss; for Set Transformer(+POT) and DeepSets(+POT), the summary network, defined as in Equations 1 and 7, is optimized by both the POT loss and task-specific loss. More experimental details are provided in Appendix C.

**Amortized Clustering with Mixture of Gaussians (MoGs):** We consider the task of maximum likelihood of MoGs with $C$ components, denoted as $P(x; \boldsymbol{\theta}) = \sum_{c=1}^{C} \pi_c N\left(x \mid \mu_c, \text{diag}\left(\sigma_c^2\right)\right)$. Given the dataset $X = \{x_{1:n}\}$ generated from the MoG, the goal is to train a neural network, which takes $X$ as input set and outputs parameters $\boldsymbol{\theta} = \{\pi_c, \mu_c, \sigma_c\}_{1,C}$. Each dataset contains $n \in [100, 500]$ points on a 2D plane, each of which is sampled from one of $C$ Gaussians. Table 1 reports the test average likelihood of different models with varying $C \in \{4, 8\}$, where we set $K = 50$ prototypes for all $C$. We observe that Set Transformer outperforms DeepSets largely, validating the effectiveness of attention mechanisms in this task. We note both Set Transformer(+POT) and DeepSets(+POT) improve their baselines, showing that the POT loss can encourage the summary networks to learn more efficient summary statistics.

**Point Cloud Classification:** Here, we evaluate our method on the task of point cloud classification using the ModelNet40 (Chang et al., 2015) dataset [1], which consists of 3D objects from 40 different categories. By treating each object as a point cloud, we represent it as a set of $N$ vectors in $\mathbb{R}^3$ (x; y; z-coordinates). Table 1 reports the classification accuracy, where we perform the experiments with varying $N \in \{64, 1024\}$, set $K = 40$ prototypes. Clearly, both DeepSets and Set Transformer can be improved by adding the POT loss. Notably, fewer points would lead to lower performance, where the POT loss plays a more important role. Taking this task as the example, we further study our model's sensitivity to hyper-parameter $\epsilon$ in Fig. 4 of Appendix C.5.

**Sum of Digits:** Following Zaheer et al. (2017), we aim to compute the sum of a given set of digits, where we consider MNIST8m (Loosli et al., 2007), consisting of 8 million instances of $28 \times 28$ grey-scale stamps of digits in $\{0, ..., 9\}$. By randomly sampling a subset of maximum $M = 10$ images from MNIST8m, we build $N = 100k$ "sets" of training, where we denote the sum of digits in that set as the set-label. We construct $100k$ sets of test MNIST digits, where we vary the $M$ starting from 10 all the way up to 100. The output of the summary network is a scalar, predicting the sum of $M$ digits. In this case, we adopt L1 as the task-specific loss and set $K = 10$ prototypes. We show the accuracy of digit summation for different algorithms in Fig. 2 and find that the POT loss can enhance the summary networks to achieve better generalization. In this task, we also explore the convergence rate and the learned transport plan matrix of Sinkhorn algorithm in Fig. 5 of the Appendix C.6.

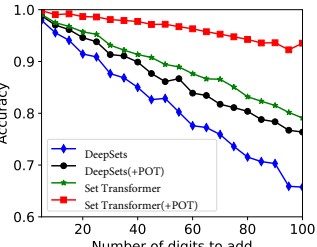

Figure 2: Accuracy of digit summation with image inputs, where all models are trained on tasks of length 10 at most and tested on examples of length up to 100.

### 5.2 EXPERIMENTS ON FEW-SHOT CLASSIFICATION

To explore whether our proposed method can improve the metric-based few-shot classification, we consider two commonly-used algorithms as the baselines, including ProtoNet (Snell et al., 2017)

---

[1] We adopt the point-cloud dataset directly from the authors of Zaheer et al. 2017

Table 1: Test performance of different methods, where left table denotes the likelihood for MoG with varying $C$ (number of components), oracle is the likelihood of true parameters for the test data, and right denotes the test accuracy for point cloud classification task with varying $N$ (number of points).

| Task | Test likelihood for MoG | | | | | Test accuracy for the point cloud classification | | | | |
|---|---|---|---|---|---|---|---|---|---|---|
| Algorithm | C=4 | C=5 | C=6 | C=7 | C=8 | N=64 | N=128 | N=256 | N=512 | N=1024 |
| Oracle | -1.473 | -1.660 | -1.820 | -1.946 | -2.058 | - | - | - | - | - |
| DeepSets | -1.809 | -1.812 | -1.897 | -2.115 | -2.261 | 79.14 | 82.51 | 84.62 | 85.74 | 86.83 |
| | ±0.015 | ±0.016 | ±0.017 | ±0.016 | ±0.014 | ±0.035 | ±0.028 | ±0.037 | ±0.045 | ±0.042 |
| DeepSets(+POT) | **-1.723** | **-1.743** | **-1.861** | **-2.078** | **-2.214** | **79.91** | **83.65** | **85.22** | **86.25** | **86.93** |
| | ±0.015 | ±0.017 | ±0.012 | ±0.018 | ±0.015 | ±0.050 | ±0.060 | ±0.055 | ±0.067 | ±0.075 |
| Set Transformer | -1.501 | -1.721 | -1.859 | -2.003 | -2.106 | 79.01 | 82.31 | 84.46 | 85.82 | 86.34 |
| | ±0.006 | ±0.006 | ±0.007 | ±0.007 | ±0.007 | ±0.103 | ±0.117 | ±0.125 | ±0.114 | ±0.122 |
| Set Transformer(+POT) | **-1.486** | **-1.676** | **-1.828** | **-1.967** | **-2.084** | **80.00** | **83.32** | **85.64** | **86.51** | **86.84** |
| | ± 0.007 | ± 0.007 | ± 0.006 | ±0.007 | ±0.007 | ±0.111 | ±0.130 | ±0.121 | ±0.124 | ±0.115 |

and MatchNet (Vinyals et al., 2016). Denoting the feature extractor in each algorithm as $f_{\phi_1}$, we consider several popular backbones, including ResNet10 and ResNet34 (He et al., 2016). Recalling the discussions in Section 3.2, to enforce $f_{\phi_1}$ to learn more powerful image features, we additionally introduce matrix $\mathbf{B}$ and net $g_{\phi_2}$ and learn the model by minimizing the POT loss and classification errors. We perform the experiments on the CUB (Welinder et al., 2010) and miniImageNet (Ravi & Larochelle, 2016). As a fine-grained few-shot classification benchmark, CUB contains 200 different classes of birds with a total of $11, 788$ images of size $84 \times 84 \times 3$, where we split the dataset into 100 base classes, 50 validation classes, and 50 novel classes following Chen et al. (2019). miniImageNet is derived from ILSVRC-12 dataset (Russakovsky et al., 2015), consisting of $84 \times 84 \times 3$ images from 100 classes with 600 random samples in each class. We follow the splits used in previous work (Ravi & Larochelle, 2016), which splits the dataset into 64 base classes, 16 validation classes, and 20 novel classes. Table 2 reports the 5way5shot and 5way10shot classification results of different methods on miniImageNet and CUB. We see that introducing the POT loss and summary network can consistently improve over baseline classifiers, and the performance gain gradually increases with the development of number of network layers. This suggests that our proposed plug-and-play framework can be flexibly used to enhance the metric-based few-shot classification, without the requirement of designing complicated models on purpose.

Table 2: 5way5shot and 5way10shot classification accuracy (%) on CUB and miniImageNet, respectively, based on 1000 random trials. Here, $(\cdot)$ is the $p$-value computed with two-sample $t$-test, and $p$-value with blue (red) color means the increase (reduce) of performance when introducing POT loss.

| Datasets | CUB | | miniImageNet | |
|---|---|---|---|---|
| ProtoNet(resnet10) | 84.32 ± 0.51 | 87.41 ± 0.49 | 72.74±0.63 | 78.14 ± 0.56 |
| ProtoNet(+OT) (resnet10) | **84.44** ±**0.51** (1e-7) | **87.69** ±**0.53** (1e-33) | **72.94** ±**0.66** (1e-12) | **78.76 ± 0.49** (1e-146) |
| ProtoNet(resnet34) | 87.33 ± 0.48 | 91.75 ± 0.47 | 73.99 ± 0.64 | 78.64 ± 0.56 |
| ProtoNet(+OT) (resnet34) | **88.34**± **0.46** (0.0) | **92.17 ± 0.48** (1e-79) | **75.15**± **0.63**(1e-265) | **79.05 ± 0.52** (1e-60) |
| MatchNet (resnet10) | 82.98± 0.56 | 85.97± 0.53 | 68.82 ± 0.65 | **72.06 ± 0.54** |
| MatchNet(+OT) (resnet10) | **83.64 ± 0.58** (1e-127) | **86.02**± **0.56** (0.04) | **68.95 ± 0.62**(1e-6) | 71.94 ± 0.56 (1e-6) |
| MatchNet (resnet34) | 84.66± 0.55 | 86.32 ± 0.56 | 68.32 ± 0.66 | **72.41 ± 0.63** |
| MatchNet(+OT) (resnet34) | **85.50 ± 0.66** (1e-172) | **86.75 ± 0.61** (1e-57) | **68.51 ± 0.64** (1e-11) | 71.98 ± 0.59 (1e-53) |

## 5.3 EXPERIMENTS ON FEW-SHOT GENERATION

Here, we consider few-shot generation task to investigate the effectiveness of our proposed implicit meta generative framework, where we consider CGAN (Mirza & Osindero, 2014) and DAGAN (Antoniou et al., 2017) as baselines for their ability to generate conditional samples. For CGAN-based models, we adopt summary network as the encoder to extract the feature vector $\boldsymbol{h}_j$ from a given set $D_j$, which is further fed into the generator and discriminator as the conditional information. Since the original DAGAN only assimilates one image into the generator, in our framework, we replace the encoder in DAGAN with summary network to learn the set representation. For DAGAN-based models, we adopt the same way with the original DAGAN to construct the real/samples for the critic and explain here for clarity: we sample two sets $(D_{1j}, D_{2j})$ from the same distribution or category; then we represent the real samples as the combination of $D_{1j}$ and $D_{2j}$ and the fake ones as the combination of $D_{1j}$ and $\hat{D}_{1j}$ from the generator (conditioned on $D_{1j}$). Different from our framework that separately optimizes the summary network using the OT loss, all other models for comparison in this paper jointly optimize the encoder (e.g., summary network) with the generator by the generator loss. Besides, for a fair comparison, we also consider introducing the additional reconstruction loss (mean square error, MSE) to optimize the generator and encoder in baselines. We

consider the DeepSets as the summary network for its simple architecture. For 3D natural images, we adopt the pretrained densenet (Iandola et al., 2014) to extract features from each data pints, and take the features as the input to summary network. To evaluate the quality of generated samples, we adopt commonly used metric Fréchet Inception Distance (FID) (Heusel et al., 2017), where we only report the FID score (Heusel et al., 2017) considering the notable performance gap between our model and the compared ones. We provide several examples on toy datasets to show the efficiency of our proposed model in Appendix E.

Table 3: FID $\downarrow$ of images generated by different methods with varying unseen angles $A$ on MNIST.

| Algorithms | A=-160 | A=-120 | A=-80 | A=-40 | A=0 | A=40 | A=80 | A=120 | A=160 |
|---|---|---|---|---|---|---|---|---|---|
| CGAN | 227.94 ± 2.56 | 203.59 ±2.21 | 211.74 ±1.88 | 231.63 ± 2.11 | 228.35 ±1.94 | 222.87 ± 2.02 | 195.30 ±1.58 | 202.69 ±1.75 | 202.35 ±1.16 |
| CGAN+MSE | 225.56 ± 2.05 | 201.09 ± 1.62 | 209.11 ±0.91 | 222.54 ± 2.13 | 223.61 ±1.55 | 220.18 ±1.47 | 193.75 ±1.39 | 200.12 ±1.08 | 201.13 ± 2.33 |
| CGAN+POT | **213.30** ± 1.35 | **193.42** ± 1.25 | **196.56** ± 1.61 | **217.28** ± 1.58 | **210.27** ± 2.04 | **206.44** ±1.78 | **181.05** ± 0.99 | **190.19** ±1.22 | **191.22** ±1.37 |
| DAGAN | 169.85 ± 1.56 | 201.58 ±1.98 | 157.95 ±1.45 | 204.23 ±2.31 | 218.42 ±1.96 | 196.70 ±1.75 | 160.35 ±1.57 | 198.24 ±2.14 | 164.64 ±1.85 |
| DAGAN+MSE | 169.77 ±1.51 | 200.12 ±2.10 | 155.87 ±1.41 | 202.16 ±1.89 | 215.12 ±1.97 | 194.61 ±2.10 | 159.08 ±1.74 | 197.48 ±2.05 | 163.23 ±1.84 |
| DAGAN+ POT | **159.51** ± 1.57 | **174.07** ±1.63 | **135.89** ±1.17 | **177.64** ±1.88 | **197.09** ±1.46 | **186.17** ±1.90 | **142.40** ±1.33 | **174.75** ±1.59 | **146.69** ±1.47 |

**Rotated MNIST:** Following Wu et al. (2020), we artificially transform each image in MNIST dataset (LeCun, 1998) with 18 rotations ($-180$ to $180$ by 20 degrees), leading to $18$ distributions characterized by angle $A$. We choose 9 interleaved distributions for training and the rest as unseen distributions for testing. We consider the CGAN-based and DAGAN-based models, respectively. During the test stage, for each unseen distribution, we randomly sample 1000 real images and generate 20 fake images based on every 20 real images and repeat this process 50 times ($i.e.$, $1000/20$), resulting in 1000 generated samples for each method. We summarize the test performance in Table 3 with varying $A$. We can find that our proposed framework allows for better generalization to related but unseen distributions at test time, indicating the POT loss can enforce the summary network to capture more salient characteristics.

**Natural Images:** We further consider few-shot image generation on Flowers (Nilsback & Zisserman, 2008) and Animal Faces (Deng et al., 2009), where we follow seen/unseen split provided in Liu et al. (2019). Flowers dataset contains 8189 images of 102 categories, which are divided into 85 training seen and 17 testing unseen categories; Animal Faces dataset contains $117,574$ animal faces collected from 149 carnivorous animal categories, which are split into 119 training seen and 30 testing unseen categories. We present the example images generated by DAGAN and DAGAN(+POT) and network architectures in Appendix F for the limited space, where we also compute the FID scores with the similar way in Rotated MNIST experiment. We find that our method achieves the lowest FID and has the ability to generate more realistic natural images compared with baselines. This indicates the summary network in our proposed framework can successfully capture the important summary statistics within the set, beneficial for the few-shot image generation.

## 6 CONCLUSION

In this paper, we present a novel method to improve existing summary networks designed for set-structured input based on optimal transport, where a set is endowed with two distributions: one is the empirical distribution over the data points, and another is the distribution over the learnable global prototypes. Moreover, we use the summary network to encode input set as the prototype proportion ($i.e.$, set representation) for global centers in corresponding set. To learn the distribution over global prototypes and summary network, we minimize the prototype-oriented OT loss between two distributions in terms of the defined cost function. Only additionally introducing the acceptable parameters, our proposed model provides a natural and unsupervised way to improve the summary network. In addition to the set-input problems, our plug-and-play framework has shown appealing properties that can be applied to many meta-learning tasks, where we consider the cases of metric-based few-shot classification and implicit meta generative modeling. Extensive experiments have been conducted, showing that our proposed framework achieves state-of-the-art performance on both improving existing summary networks and meta-learning models for set-input problems. Due to the flexibility and simplicity of our proposed framework, there are still some exciting extensions. For example, an interesting future work would be to apply our method into approximate Bayesian computation for posterior inference.

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

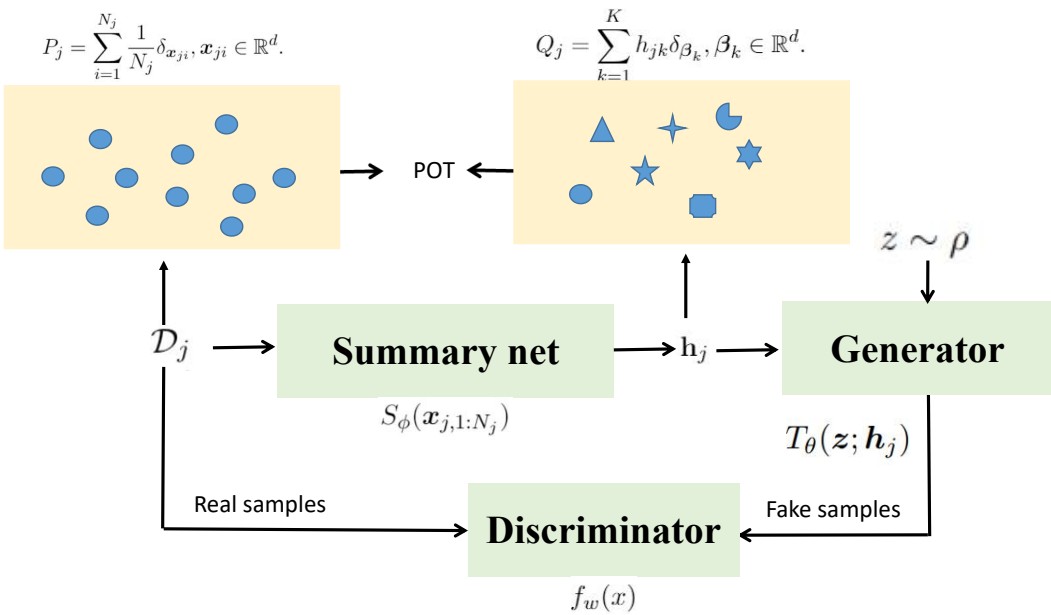

$$P_j = \sum_{i=1}^{N_j} \frac{1}{N_j} \delta_{\boldsymbol{x}_{ji}}, \boldsymbol{x}_{ji} \in \mathbb{R}^d. \qquad Q_j = \sum_{k=1}^{K} h_{jk} \delta_{\boldsymbol{\beta}_k}, \boldsymbol{\beta}_k \in \mathbb{R}^d.$$

Figure 3: The overview of our proposed implicit meta generative framework, where we sample the $j$-th distribution from training sets, feed the data points into the summary network, generate the fake samples with the random noise and summary output as the input, discriminate the real/fake samples.

---

**Algorithm 2** The workflow of our proposed implicit meta generative framework.

---

**Require:** Datasets $\mathcal{D}_{1:J}$, initial discriminator parameters $w$, initial generator parameters $\theta$, initial summary network parameters $\phi$, initial matrix $\mathbf{B}$, the cost function $C$, the number of critic iterations per generator iteration $\eta_{critic}$, the batch size $m$, learning rate $\alpha$ and the hyper-parameter $\epsilon$.

**while** $w, \theta, \mathbf{B}, \phi$ has not converged **do**
  Randomly choose $j$ from $1, 2, .., \mathcal{J}$
  **for** $t = 1, \cdots, \eta_{critic}$ **do**
    Sample the real data set $D_j = \{\boldsymbol{x}_{ji}\}_{i=1}^{m}$ from $j$-th empirical distribution $P_j$;
    Embed the observed batch data into the statistics $\boldsymbol{h}_j = \text{Softmax}(S_\phi(D_j))$;
    Sample a batch of prior samples $\{z_i\}_{i=1}^{m}$ from $p(z)$;
    $g_w \leftarrow -\nabla_w \frac{1}{m} \sum_{i=1}^{m} [\log f_w(x_j^i) + \log(1 - f_w(T_\theta(z_i, \boldsymbol{h}_j)))]$;
    $w \leftarrow w + \alpha g_w$
  **end for**;
  Represent the $Q_j$ with global prototype matrix $\mathbf{B}$ and set representation/statistics $\boldsymbol{h}_j$
  Compute the loss $\text{OT}_\epsilon(P_j, Q_j)$ between $P_j$ and $Q_j$ with Sinkhorn algorithm in Equation 6
  $g_{\mathbf{B}} \leftarrow \nabla_{\mathbf{B}} [\text{OT}_\epsilon(P_j, Q_j)]$;
  $\mathbf{B} \leftarrow \mathbf{B} + \alpha g_{\mathbf{B}}$;
  $g_\phi \leftarrow \nabla_\phi [\text{OT}_\epsilon(P_j, Q_j)]$;
  $\phi \leftarrow \phi + \alpha g_\phi$; see Algorithm 1 for more details;
  Sample a batch of latent variables $\{z^{(i)}\}_{i=1}^{m} \sim p(z)$;
  $g_\theta \leftarrow \nabla_\theta \left[ \frac{1}{m} \sum_{i=1}^{m} \log(1 - f_w(T_\theta(z_i, \boldsymbol{h}_j))) \right]$;
  $\theta \leftarrow \theta + \alpha g_\theta$;
**end while**

---

# A ALGORITHMS AND ILLUSTRATION OF OUR PROPOSED MODEL

The pseudo code for the implicit meta generative modeling is provided in Algorithm 2.

## B THE DIFFERENCE BETWEEN OUR MODEL AND BARYCENTER PROBLEM

In this section, we clarify the difference between Wasserstein barycenter and our method. Specifically, for $j$-th distribution, we denoted $P_j$ as its empirical distribution consisting of $N_j$ samples, expressed as $P_j = \sum_{i=1}^{N_j} \frac{1}{N_j} \delta_{\boldsymbol{x}_{ji}}, \boldsymbol{x}_{ji} \in \mathbb{R}^d$. Notably, $\boldsymbol{a}_j = [\frac{1}{N_j}] \in \Sigma^{N_j}$ represents the probability measure for distribution $P_j$.

For another thing, we can represent $P_j$ with another to-be-learned distribution $Q_j$, defined as $Q_j = \sum_{k=1}^{K} h_{jk} \delta_{\boldsymbol{\beta}_k}, \boldsymbol{\beta}_k \in \mathbb{R}^d$. Here $\boldsymbol{b}_j = [h_{jk}] \in \Sigma^K$ is the probability measure for distribution $Q_j$, which can be computed using summary network $S_\phi(\boldsymbol{x}_{i,1:N_j})$ and serves as the representation for set $j$. And $\boldsymbol{\beta}_k$ is the $k$-th prototype in the same space of the observed data points, which is the $k$-th column of $\mathbf{B} \in \mathbb{R}^{K \times d}$, a learnable global prototype matrix. To optimize the $\boldsymbol{\beta}_{1:K}$ and the summary network $S_\phi$ for computing $\boldsymbol{h}_j$, we minimize the average OT loss (between $Q_j$ and $P_j$) for all training sets. We rewrite the Equation (6) here for convenience:

$$L_{\mathrm{OT}} = \min_{\mathbf{B}, \phi} \frac{1}{\mathcal{J}} \sum_{j=1}^{\mathcal{J}} \left( \sum_{i}^{N_j} \sum_{k}^{K} C_{ik} T_{ik} - \epsilon \sum_{i}^{N_j} \sum_{k}^{K} -T_{ik} \ln T_{ik} \right) = \min_{\mathbf{B}, \phi} \frac{1}{\mathcal{J}} \sum_{j=1}^{\mathcal{J}} \left( \mathrm{OT}_\epsilon(P_j, Q_j) \right).$$

Usually, this equation can also be represented:

$$\min_{\mathbf{B}, \phi} \frac{1}{\mathcal{J}} \sum_{j=1}^{\mathcal{J}} \left( \mathrm{OT}_\epsilon(\boldsymbol{a}_j, \boldsymbol{b}_j) \right). \tag{10}$$

In terms of Wasserstein barycenter, we adopt the same notations for consistency. Following (Altschuler & Boix-Adserà, 2021), given $J$ empirical distributions $P_{1:J}$ and their respective probability measures $[\boldsymbol{a}_1, \ldots, \boldsymbol{a}_J]$ supported on $\mathbb{R}^d$ and a vector $\boldsymbol{\lambda} \in \Sigma^J$, their corresponding Wasserstein barycenter can be viewed as another distribution $Q$, i.e., $Q = \sum_{k=1}^{K} m_k \delta_{\boldsymbol{\beta}_k}, \boldsymbol{\beta}_k \in \mathbb{R}^d$, where $\boldsymbol{\beta}_k$ is the $k$-th column of $\mathbf{B}$, $\boldsymbol{m} = [m_k] \in \Sigma^K$ is the probability measure for distribution $Q$. Then we can learn the barycenter (i.e., $\mathbf{B}$ and $\boldsymbol{m}$) by minimizing

$$\min_{\mathbf{B}, \boldsymbol{m}} \frac{1}{\mathcal{J}} \sum_{j=1}^{\mathcal{J}} \lambda_j \mathcal{W}(P_j, Q) = \min_{\mathbf{B}, \boldsymbol{m}} \frac{1}{\mathcal{J}} \sum_{j=1}^{\mathcal{J}} \lambda_j \mathcal{W}(\boldsymbol{a}_j, \boldsymbol{m}) \tag{11}$$

where above $\mathcal{W}(\cdot, \cdot)$ denotes the squared 2-Wasserstein distance. By comparing the equation 11 and equation 10, we can find that our model learns a $Q_j$ to approximate $P_j$ for each distribution $j$ but "barycenter" problem learns a shared $Q$ as the barycenter for all $P_{1:J}$. Therefore, after minimizing the average loss for all training sets with Equation (6), we can use the probability measure $\boldsymbol{b}_j = [h_{jk}]$ (i.e., $\boldsymbol{h}_j$) to represent the empirical distribution $P_j$. Especially, we can directly map the test set to its representation by using the summary network. However, since the probability measure $\boldsymbol{m}$ in "barycenter" is shared by all distributions, it can not represent a specific set. Therefore, to achieve the set representation, it might need to first compute the transport plan between test set and the barycenter $Q$ and then aggregate the data points (or features) within the test set by taking the transport plan as the weight. Therefore, our model produces a more intuitive solution to learn the set representation, which can take full advantage of the existing summary networks and provides a promising tool for addressing set-input and meta-learning problems.

## C EXPERIMENTAL SETTINGS ABOUT INTRODUCING POT LOSS INTO THE SUMMARY NETWORKS

### C.1 DETAILS FOR AMORTIZED CLUSTERING WITH MIXTURES OF GAUSSIANS

We generate the 2D toy datasets following the Lee et al. (2019), where we additionally vary the $C$ (the number of components) from 4 to 8. Below, we present the detailed generation process about the toy datasets:

1. Specify the number of components $C$ for 2D toy dataset.
2. Generate the number of data points, $n \sim \mathrm{Uniform}(100, 500)$.
3. Sample the mean vector for $C$ components.

$$\mu_{c,d} \sim \mathrm{Uniform}(-4, 4), \quad c = 1, \ldots, C, \quad d = 1, 2$$

4. Sample the cluster labels.

$$\pi \sim \text{Dir}\left(\mathbf{1}^{\top}\right), \quad z_i \sim \text{Categorical}(\pi), i = 1, \ldots, n, z_i = 1, \ldots, C$$

5. Generate data from spherical Gaussian.

$$x_i \sim \mathcal{N}\left(\mu_{z_i}, (0.3)^2 I\right), i = 1, \ldots, n$$

## C.2 Details about Set-Transformer-based and DeepSets-based architectures used in MoGs experiments

**DeepSets** In terms of the DeepSets, the $f_{\phi_1}$ in summary network contains 3 permutation-equivariant layers with 256 channels followed by mean-pooling over the set structure. Then the resulting vector representation $z_j$ of the set is then fed to a fully connected layer with 512 units followed by a linear layer $512 \times C(1 + 2 * 2)$, where $C$ denotes the number of components. We use ELU activation at all layers. To introduce the POT loss into the DeepSets, we further feed the $z_j$ into a fully connected layer with 512 units followed by a 50-way softmax unit and also introduce the global matrix $\mathbf{B} \in \mathbb{R}^{2 \times 50}$.

**Set Transformer** To perform the MoGs experiments, we adopt the same architecture for Set Transformer following Lee et al. (2019), whose parameters are reported in Table 4. To introduce the POT loss, we also add the two fully connected layers with 256 units on the resulting vector $z_j$ followed by a 50-way softmax unit, and a global prototype matrix $\mathbf{B} \in \mathbb{R}^{2 \times 50}$.

## C.3 Details about Set-Transformer-based and DeepSets-based architectures used in sum of digits

**DeepSets** Following the official code in Zaheer et al. (2017), we adopt the default architecture to implement the DeepSets, where we first project the image into a 128-dimensional vector with three convolutional layers and apply summary network on the 128-dimensional vectors. To build DeepSets(+POT), we take the set representation $z_j$ after sum-pooling in summary network as the input and introduce fully connected layer with 128 units followed by a 10-way softmax unit, and a global prototype matrix $\mathbf{B} \in \mathbb{R}^{128 \times 10}$.

**Set Transformer** For Set Transformer, we follow the similar structure used in MoGs experiments, where we also project the image with three convolutional layers and output a scalar. To build Set Transformer(+POT), we take the representation $z_j$ after sum-pooling in summary network as the input and introduce fully connected layer with 128 units followed by a 10-way softmax unit, and a center matrix $\mathbf{B} \in \mathbb{R}^{128 \times 10}$.

## C.4 Details about Set-Transformer-based and DeepSets-based architectures used in point cloud classification

**DeepSets** For original DeepSets, we adopt the same architecture with Zaheer et al. (2017). In a specific, the $f_{\phi_1}$ in summary network contains 3 permutation-equivariant layers with 256 channels followed by max-pooling over the set structure. Then the resulting vector representation $z_j$ of the set is then fed to a fully connected layer with 256 units followed by a 40-way softmax unit. We use Tanh activation at all layers and dropout on the layers after set-max-pooling (*i.e.*, two dropout operations) with 50% dropout rate. To introduce the POT loss into the DeepSets, we further feed the $z_j$ into a fully connected layer with 256 units followed by a 40-way softmax unit, with 70% dropout rate. Besides, we additionally introduce the center matrix $\mathbf{B} \in \mathbb{R}^{3 \times 40}$.

**Set Transformer** We also adopt the same architecture to implement the Set Transformer, where we summarize the parameters in Table 5, following Lee et al. (2019). To improve the Set Transformer with POT loss, we also introduce a fully connected layer with 256 units followed by a 40-way softmax unit, with 90% dropout rate, and a center matrix $\mathbf{B} \in \mathbb{R}^{3 \times 40}$.

## C.5 Parameter sensitivity

In the previous experiments, we fix the value of $\epsilon$ as 0.1, controlling the weight of the entropic regularisation in the Sinkhorn algorithm. Notably, unless specified otherwise, we specify the construction of $\mathbf{C}$ as $C_{ik} = 1 - \cos\left(\boldsymbol{x}_{ji}, \boldsymbol{\beta}_k\right)$. Therefore, the cost function provides an upper-bounded

Table 4: Detailed architectures of Set Transformer used in the MoGs experiments, cited from Lee et al. (2019), where $C$ denotes the number of components.

| Encoder | | | Decoder | |
|---|---|---|---|---|
| **rFF** | **SAB** | **ISAB** | **Pooling** | **PMA** |
| FC(128, ReLU) | SAB(128, 4) | ISABm(128, 4) | mean | PMA4(128, 4) |
| FC(128, ReLU) | SAB(128, 4) | ISABm(128, 4) | FC(128, ReLU) | SAB(128, 4) |
| FC(128, ReLU) | - | - | FC(128, ReLU) | FC(C · (1 + 2 · 2), ) |
| FC(128, ReLU) | - | - | FC(128, ReLU) | FC(C · (1 + 2 · 2), ) |
| - | - | - | FC(C · (1 + 2 · 2), ) | - |

Table 5: Detailed architectures of Set Transformer used in the point cloud classification experiments, cited from Lee et al. (2019).

| Encoder | | Decoder | |
|---|---|---|---|
| **rFF** | **ISAB** | **Pooling** | **PMA** |
| FC(256, ReLU) | ISAB(256, 4) | max | Dropout(0.5) |
| FC(256, ReLU) | ISAB(256, 4) | Dropout(0.5) | PMA1(256, 4) |
| FC(256, ReLU) | - | FC(256, ReLU) | Dropout(0.5) |
| FC(256, ) | - | Dropout(0.5) | FC(40, ) |
| - | - | FC(40, ) | - |

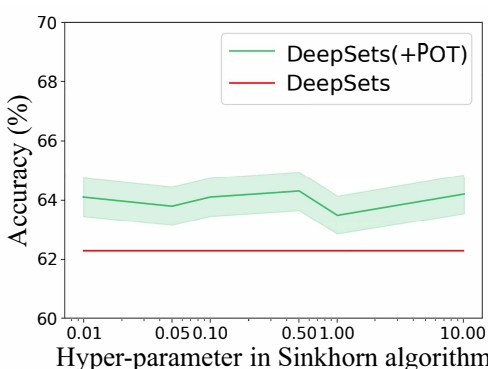

Figure 4: Parameter sensitivity of DeepSets(+POT) on point cloud classification task, with varying $\epsilon$, where each object is represented as a set of $N = 20$ vectors.

positive similarity metric, making the $\epsilon$ has the corresponding reasonable range as a prior knowledge. Here, we study our DeepSets(+POT)'s sensitivity to $\epsilon$. We consider the point cloud classification task and each object is represented as a set of $N = 20$ vectors. As shown in Fig. 4, we report the performance of DeepSets(+POT) on point cloud classification task with varying $\epsilon$, where DeepSets serves as the baseline. It can be seen that our model is robust to the $\epsilon$. Besides, all the results of DeeepSets(+POT) with different $\epsilon$ are superior than that of DeeepSets, indicating the effectiveness of our method. By fine-tuning $\epsilon$ for each dataset in each task, we might obtain better results than those reported in our experiments. However, we aim to validate our method instead of exhaustively tuning this hyper-parameter and thus we set $\epsilon = 0.1$, which can achieve the acceptable result.

## C.6 Convergence rate of Sinkhorn algorithm

In this paper, we set the maximum iteration number as Itermax $= 200$ in Sinkhorn algorithm for all experiments. As shown in Fig. 5, we visualize the convergence rate of Sinkhorn algorithm, where we consider the task about "sum of digits" (DeepSets+POT). The upper figure shows the convergence rate of Sinkhorn algorithm. The bottom figure visualizes the transport plan matrix with varying iterations. We find that the 200 iterations are typically enough for Sinkhorn algorithm and we can learn a sparse transport plan matrix $\mathbf{T}$ when the algorithm converge. Notably, the transport plan matrix needs to satisfy two marginal constraints, defined by the probability measures of two distributions, respectively. Recall that the empirical distribution has an unchanged uniform probability measure, so the learned transport plan matrix is dense for the $N_j$ observed samples. In terms of another distribution, its probability measure is the set-specific representation, weighting the

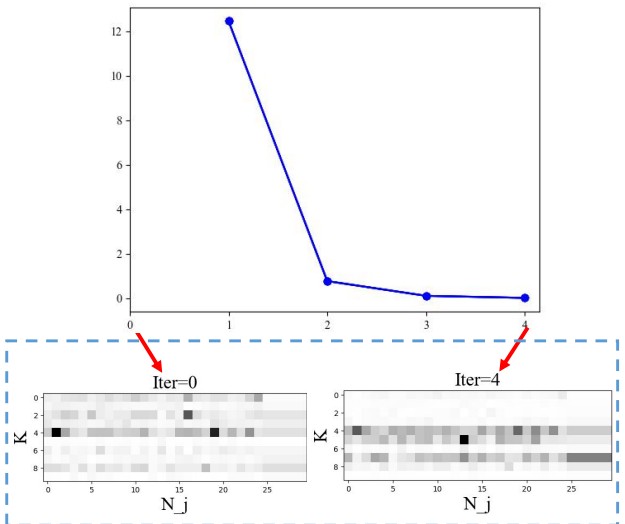

Figure 5: Top: the convergence rate of Sinkhorn for $c(x, y) = 1 - cosine(x, y)$, and $\epsilon = 0.1$, as measured in term of marginal constraint violation $\frac{1}{J} \sum_j^J \sum_i^{N_j} |u_{ij}^{l+1} - u_{ij}^l|$, where $l$ is the iteration index and $\boldsymbol{u}$ is the scaling variable; please see page 67 in Peyré & Cuturi (2019) for more details. Bottom: evolution of the transport plan matrix $\mathbf{T} = \mathrm{diag}\left(\boldsymbol{u}^{(\ell)}\right) \mathbf{K} \, \mathrm{diag}\left(\boldsymbol{v}^{(\ell)}\right)$ computed at iteration of Sinkhorn's iterations.

importance of K shared centers for corresponding set. Therefore, it is reasonable that transport plan matrix is sparse for K centers.

# D    EXPERIMENTAL SETTINGS ABOUT FEW-SHOT CLASSIFICATION

Denote the prototype for set $j$ (computed by $f_{\phi_1}$) in few-shot classification as $\boldsymbol{c}_j$. We consider two backbones for $f_{\phi_1}$, including ResNet10 and ResNet34, which produce the 512-dimensional $\boldsymbol{c}_j$. To improve the metric-based few-shot classification with our framework, taking the $\boldsymbol{c}_j$ as input, we further construct the $g_{\phi_2}$. Specifically, we introduce a fully connected network with architecture as $512 \rightarrow 256$ units with ReLU function followed by a $\mathbf{X}$-way ($\mathbf{X}$=64 for CUB, and $\mathbf{X}$=128 for miniImageNet) Softmax function and a center matrix $\mathbf{B} \in \mathbb{R}^{512 \times \mathbf{X}}$. We conduct 10000 tasks of the training set $\mathcal{D}_{tr}$ to train the model while 1000 tasks of the test set $\mathcal{D}_{te}$ to evaluate the learned model. And $\mathcal{D}_{tr} \cap \mathcal{D}_{te} = \emptyset$. We run 60 epochs to train the model on CUB and miniImageNet. The model is trained using Adam optimizer with default settings (learning rate $1e-3$, $\beta = (0.9, 0.999)$, and $\epsilon = 1e-8$) on one Nvidia Geforce RTX3090 GPU.

# E    ADDITIONAL EXPERIMENTAL RESULTS ON FEW-SHOT GENERATION ABOUT TOY DATASETS

We test our algorithm through a series of synthetic data sets and realistic data sets. For synthetic datasets, we set $T_\theta$, $f_w$ and $S_\phi$ as fully connected neural networks, where $T_\theta$, $f_w$ have 4 hidden layers and $f_{\phi_1}$ and $g_{\phi_2}$ (we adopt DeepSets) have 3 hidden layers. Each layer has 200 nodes, and the activation function is chosen as RELU, where we adopt the softmax in the final layer.

**Normal distribution on 2D toy data:** We first consider the 2D normal case, where the training data contains $10K$ sets and each set contains 100 data points from $N(\boldsymbol{\mu}, \boldsymbol{\Sigma})$. We sample the mean, variance, and covariance from $U[-5, 5]$, $U[1, 2]$, and $U[-0.5, 0.5]$, respectively. Fig. 6 shows the real (gray points) and generated samples (red points) by different models given unseen test sets, where we only consider CGAN-based methods for the simple toy data. We find that our model (third column) can improve the resistance to mode collapse compared with CGAN+MSE (second column) and better fit the unseen test distributions than CGAN (first column). This result indicates the POT loss can spur the summary network to capture more desired statistics for unseen distributions.

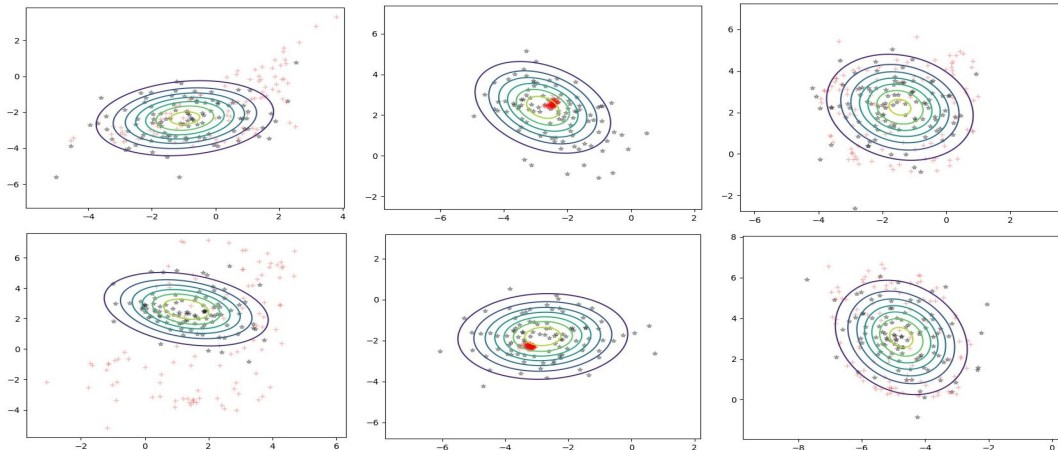

Figure 6: Examples of few-shot generation for two dimensional Gaussian distributions, where we visualize the real samples from the true distributions (gray points) and generated samples (red points) by different models (from first to third column: CGAN, CGAN+MSE and our proposed CGAN+POT), where we also plot the contour of each Gaussian distribution.

**One-dimensional Gaussian distributions:** In this case, we generate another collection of synthetic $1 - D$ datasets based on Gaussian parametric family, where the means and variances are sampled from U$[-1, 1]$ and U$[0.5, 2]$ respectively. The training data contains $10K$ sets each containing 50 samples. We also visualize the pdfs of gaussian distributions with randomly means and variance in Fig 7, which are used to sample test data, and show the 500 data points generated by the push-forward. For this experiment, we set the dimension of $z$ and summary vector $s$ as 2.

**Multi-family distribution on 1D toy data:** To validate if our proposed model can capture many types of distributional families simultaneously, we construct a collection of synthetic 1-D datasets each containing 100 samples from either an Exponential, Gaussian or Laplacian distribution with equal probability. For Gaussian and Laplacian distributions, means and variances are sampled from $U[-1, 1]$ and $U[0.5, 2]$ respectively; for Exponential distributions, rates are sampled from $U[0.5, 2]$. Fig. 8 visualizes the pdfs of six one-dimensional test distributions with different means and variances and the generated data points. It is interesting to observe that the generated data points can fit the corresponding pdf well, indicating our model can generalize to different distributions with varying parameters. Besides, our model performs slightly worse on the Exponential distributions, perhaps attributing to the fact that it is the only non-symmetric distribution.

## F    DETAILS ABOUT NATURAL IMAGE GENERATIONS AND THE RESULTS

We use denseNet proposed by Huang et al. (2018) as the backbone of summary network, then a pooling operation is conducted as Zaheer et al. (2017) does. And a 2-layer fully connected network with ReLU activation function is finally employed to embed the 4096-D visual features into the corresponding 512-D set representations $\boldsymbol{h}_j$. As for conditional generator (conditioned on set representations as well as Gaussian noise), we introduce a 2-layer embedding network [100→600→100] with LeakyReLU activation function to embed the input noise **n**. Besides, we use a 5-layer deconvolution network [ConvTranspose2d($\mathbf{X}$ + 100, 512, 4, 1, 0)→ ConvTranspose2d(512, 256, 4, 2, 1)→ConvTranspose2d(512, 128, 4, 2, 1)→ConvTranspose2d(128, 64, 4, 2, 1)→ConvTranspose2d(64, 3, 4, 2, 1)], where $\mathbf{X} = 64, 128$ for oxford and animal face datasets respectively with BatchNorm along channels and ReLU activation function to deconvolute the concatenated noise embeddings and set representations $cat([\mathbf{n}, \boldsymbol{h}_j])$ as fake output images $\boldsymbol{x}_{fake}$ : $\mathbb{R}^{N \times 3 \times 64 \times 64}$. Finally, a 5-layer discriminator network [Conv2d(2*3, 64, 4, 2, 1)→Conv2d(64, 128, 4, 2, 1))→Conv2d(128, 256, 4, 2, 1)→Conv2d(256, 512, 4, 2, 1)→Conv2d(512, 1, 4, 1, 0)] with BatchNorm as well as LeakyReLU activation function at the first fourth deconvolutional layers and the last layer without BatchNorm while with sigmoid activation funtion to distinguish the true or fake generated images. We present the generated results in Figure 9.

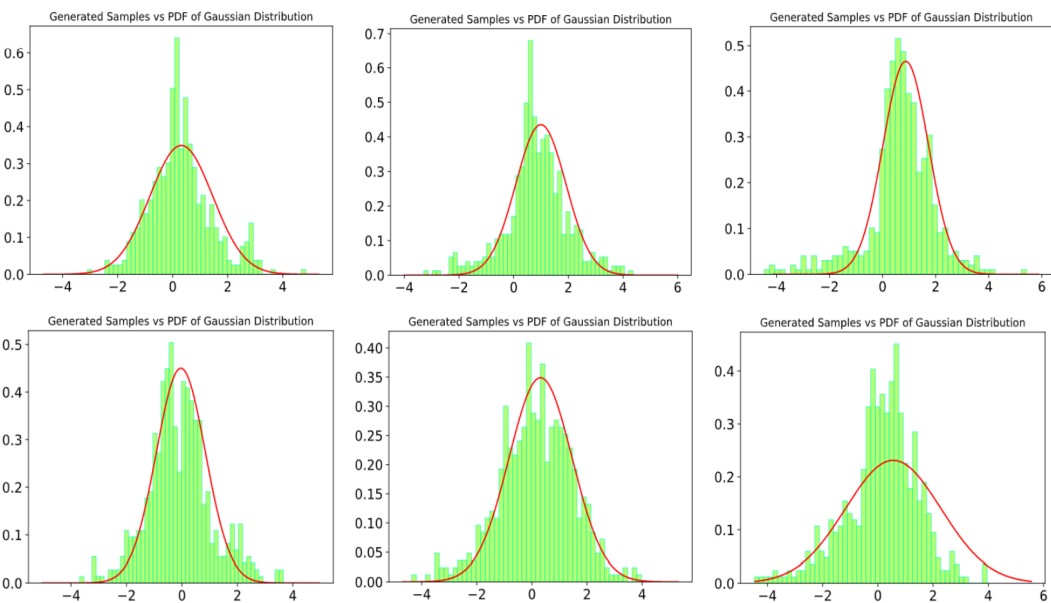

Figure 7: Examples of few-shot generation for one dimensional Gaussian distributions, where we visualize the generated samples by our GAN+POT (conditioned on the test samples from the unseen distribution) and the PDF of the unseen true distribution.

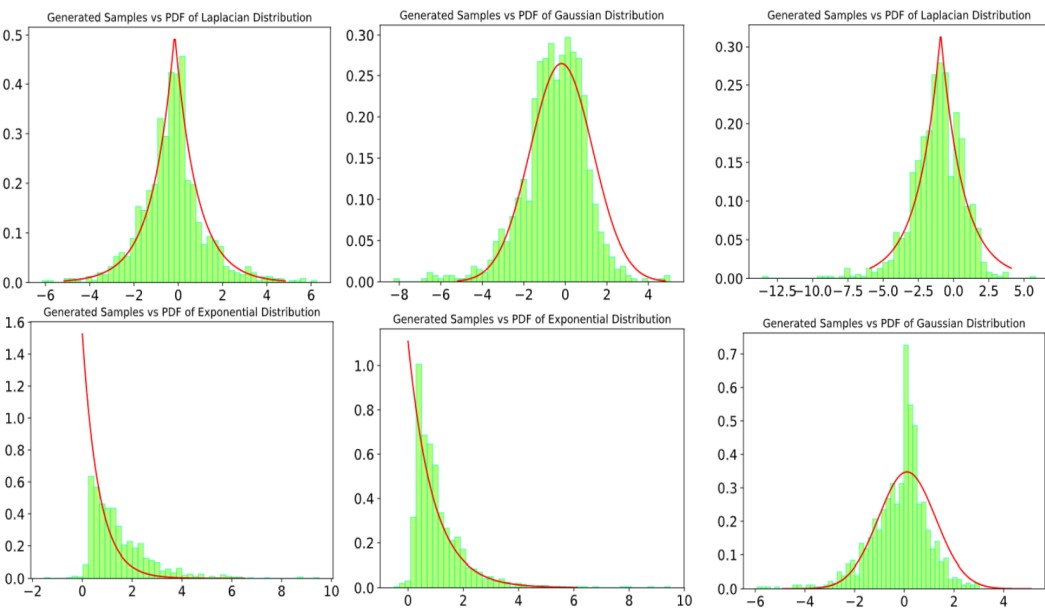

Figure 8: Examples of few-shot generation for multi-distributions, where we visualize the generated samples (green) by our GAN+POT (conditioned on the test samples from the unseen distribution) and the PDF (red) of the unseen true distribution.

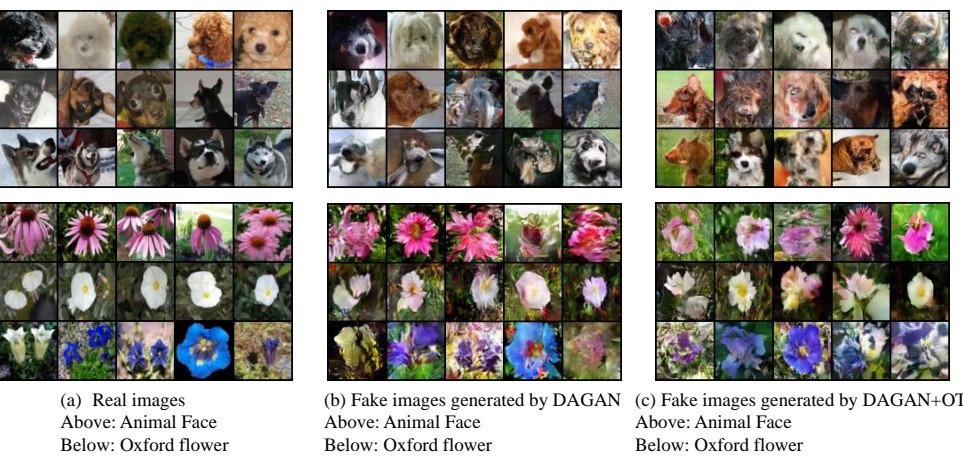

(a) Real images
Above: Animal Face
Below: Oxford flower

(b) Fake images generated by DAGAN
Above: Animal Face
Below: Oxford flower

(c) Fake images generated by DAGAN+OT
Above: Animal Face
Below: Oxford flower

Figure 9: Examples of few-shot generation for natural images, where the images are generated by DAGAN (second column) and DAGAN+POT (third column) conditioned on 3 different categories on Oxford(flower) and animal face datasets. The FID ↓ scores of DAGAN and DAGAN+POT on Oxford are 97.25 and **91.78**, respectively. The FIDs of DAGAN and DAGAN+POT on animal face are 139.14 and **131.51**.

