# OpenReview forum: "Learning Prototype-oriented Set Representations for Meta-Learning "
_ICLR.cc/2022/Conference — ICLR 2022 Poster_

### Official Review · Reviewer_AAZ1 · 2021-10-26

**Correctness:** 4
**Technical Novelty And Significance:** 2
**Empirical Novelty And Significance:** 3
**Recommendation:** 6
**Confidence:** 4

**Main Review:**

Strengths
1) This paper provides a promising research direction in using set-structured data on various downstream tasks. Especially, the paradigm using well-learnt set-representations as an conditional input deserves more explorations to tackle the limitation of few-shot generation problems.

2) The proposed algorithm is well-motivated, and is effective to be implemented for real-world applications. In particular, it is easy to combine the summary network with the majority of existing metric-based meta-learning models and generative models.

3) The experimental study is extensively conducted on various tasks related to set-structured data, and the improvement is consistent over different tasks. This shows the effectiveness of the proposed plug-and-play framework.

Weakness
1) The main idea of this paper that uses optimal transport in learning representation, has been widely studied such as https://openreview.net/pdf?id=ZK6vTvb84s, https://arxiv.org/pdf/2007.05840.pdf. It would be better to clarify what the contribution of this paper is and what the major difference is in contrast to the existing literature of optimal transport for representation learning.

**Summary Of The Paper:**

This paper proposed an improved training of summary network using optimal transport based auxiliary loss. The summary network with negligibly increased parameters show much better performance on amortized clustering, point-cloud classification, sum of digits, few-shot classification, and few-shot generation tasks.

**Summary Of The Review:**

This paper is well-motivated and definitely effective for real-world meta-learning applications, however, the authors should discuss the literature of representation learning using optimal transport.

---

> ### Author Response · Authors · 2021-11-22
> **Response to Reviewer AAZ1**
>
> Thanks for your comments. We agree with you that using optimal transport in learning representation has been studied in https://openreview.net/pdf?id=ZK6vTvb84s, and https://arxiv.org/pdf/2007.05840.pdf, referred to as [Ref1] and [Ref2], respectively. However, there are fundamental differences between theirs and ours, which have been clarified in the related work section in the revised version and are explained in detail below.
>
>
> In terms of optimal Transport Kernel Embedding (OTKE) proposed by [Ref1], it marries ideas from OT theory and kernel methods, which aligns features (embedded with kernel method) of a given set to a trainable reference distribution. To our best knowledge, OTKE views the ``reference'' as the barycenter by assuming its probability measure as a uniform measure, and uses the Sinkhorn algorithm to compute the transport plan between the shared reference and a given set. OTKE then aggregates all features in a set by viewing the transport plan as the weight, achieving the set-specific representation.  Although OTKE adopts OT theory and the elements in its reference are similar to the shared prototypes (centers) in our work, there are fundamental differences between OTKE and ours. More specifically, we construct $J$ to-be-learned distributions (rather than one reference distribution with a uniform measure) to approximate the corresponding $J$ empirical distributions, respectively. Each trainable distribution is viewed as a distribution over a set of global prototypes, whose probability measure is defined by the set-specific representation. In this paper, each set-specific representation can be directly computed with the summary network. Therefore, the summary network and shared prototypes can be directly optimized by minimizing the OT loss between the to-be-learned distributions and empirical distributions. Recall that OTKE adopts kernel-based methods to compute feature of each data point within a set and aggregates them into set-specific representation by taking the transport plan as the weight. However, we can directly encode the test set to its representation with the summary network, avoiding iteratively computing the transport plan between the test set and learned reference like OTKE. As a flexible framework, our proposed method can be naturally utilized to improve existing summary networks without changing the model architecture a lot.
>
>
> Ref [2] is also proposed to learn compact (low-dimensional) representations for sequential data with the help of OT, which however blends contrastive learning, adversarial learning, optimal transport, and Riemannian geometry into one framework. In detail, Ref [2] adopts the Wasserstein GAN to generate a set $Y$ of adversarial noise samples by taking an observed set $X$ as input. Then it uses $X$ and $Y$ with a joint optimal transport and representation learning formulation that tries to (i) minimize the optimal coupling between the two sets, while (ii) also learn a subspace $U$ that maximizes the distance between projections of $X$ onto $U$ and the adversarial noise $Y$. The idea is that the representation $U$ can filter this dimension via contrasting against the noise. However, our work directly minimizes the OT distance between empirical distribution and the corresponding to-be-learned distribution (defined by shared prototypes and set-specific representation), where there are no generative processes from the original set to the adversarial noise set and the contrastive representation learning processes. Moreover, [Ref1]  and [Ref2]  mainly focus on either biological sequence classification tasks or human action recognition in video sequences, while our model provides a promising tool for addressing not only the general set-input problems but also the meta-learning problems. These differences between Refs [1-2] and ours lead to different views of set representation learning and different frameworks as well.
>
> [Ref1]: A Trainable Optimal Transport Embedding for Feature Aggregation and its Relationship to Attention (Gregoire Mialon, Dexiong Chen, Alexandre d'Aspremont, Julien Mairal) [ICLR 2021]
> [Ref2]: Representation Learning via Adversarially-Contrastive Optimal Transport [ICML 2020]

---

### Official Review · Reviewer_tuTq · 2021-10-29

**Correctness:** 3
**Technical Novelty And Significance:** 2
**Empirical Novelty And Significance:** 2
**Recommendation:** 6
**Confidence:** 2

**Details Of Ethics Concerns:**

None.

**Main Review:**

My main concern is on the novelty of this work. Indeed, it seems to me that some ideas presented here can be found in previous papers on set representation learning that are not mentioned. In [1], prototypes against which OT is computed are learned (potentially jointly with an element-wise embedding) with or without supervision, although the outputs used in practice seem to differ. [2] presents a similar method in the context of graph representation. It is therefore difficult to me to understand the real contributions of this paper: it would be great if the authors could comment on the similarities and differences between their method and these ones.

Pros:
- The paper is well-motivated and mostly clear.
- Experiments are varied and seems sound.
- In particular, the proposed framework seems to improve existing methods for set representations (DeepSets and Set Transformer).

Cons:
- I have concerns on the novelty of this work.

Questions and remarks:
- Equation (6) seems to be an instance of Wasserstein barycenter: could you comment on this?
- It could be worth to look at this other paper on set representation in the related work [3] which has a different approach to the current and previous work yet also similar ideas.
- Could the authors elaborate on the computational complexity of computing the transport plans? How many Sinkhorn iterations do you use?
- How sensitive are you results with respect to the entropic regularization parameter $\epsilon$ ?

------------------

[1]  A Trainable Optimal Transport Embedding for Feature Aggregation and its Relationship to Attention (Grégoire Mialon, Dexiong Chen, Alexandre d'Aspremont, Julien Mairal)

[2] Wasserstein Embedding for Graph Learning (Soheil Kolouri, Navid Naderializadeh, Gustavo K. Rohde, Heiko Hoffmann)

[3] Rep the Set: Neural Networks for Learning Set Representations (Konstantinos Skianis, Giannis Nikolentzos, Stratis Limnios, Michalis Vazirgiannis)

**Summary Of The Paper:**

 This work proposes a method to improve set representation, and applies it in the context of meta-learning. More precisely, the method consists in jointly learning a summary network and prototypes using an optimal transport loss: the prototypes and summary network output should minimize the sum of the optimal transport costs for all the sets of the meta dataset. Experiments are conducted in the context of few-shot learning and other tasks used to evaluate summary networks.

**Summary Of The Review:**

The paper seems sound but I have concerns on the novelty: I am willing to raise my score if the authors clarify their contribution w.r.t [1] and [2].

---

> ### Author Response · Authors · 2021-11-22
> **Response to Reviewer tuTq (2/3)**
>
> Q2: Thanks for your reference. We agree that RepSet has a similar idea to ours and we discussed RepSet in the related work section in the revised version. More specifically, RepSet computes some comparison costs between the input sets and some to-be-learned reference sets, which is solved by using a network flow algorithm, such as bipartite matching. These costs are then used as set representation in a subsequent neural network. However, unlike our approach, RepSet does not allow unsupervised learning. As a plug-and-play module, our OT loss can serve as an additional loss for the supervised learning; it also allows the unsupervised learning to enforce the summary network for extracting powerful set representation, which makes integrating the existing summary network into the meta-learning frameworks possible. Besides, since we build the summary network as the encoder in learning set representation, we can directly map the test set into its feature without other calculations or iterations.
>
> Q3: As demonstrated by [Ref2-Ref4], to approximate the general optimal transport (OT) distance between two discrete distributions of size $n$, the time complexity bound scales as $n^{2} \log (n) / \varepsilon^{2}$ to reach $\varepsilon$-accuracy with Sinkhorn's algorithm.
> [Ref2]: Pavel Dvurechensky, Alexander Gasnikov, and Alexey Kroshnin. Computational optimal
> transport: Complexity by accelerated gradient descent is better than by Sinkhorn’s algorithm.
> In International Conference on Machine Learning, pages 1367–1376, 2018.
> [Ref3]: Jason Altschuler, Jonathan Niles-Weed, and Philippe Rigollet. Near-linear time approximation
> algorithms for optimal transport via Sinkhorn iteration. In Advances in Neural Information
> Processing Systems, pages 1964–1974, 2017.
> [Ref4]: Chizat, L., Roussillon, P., Léger, F., Vialard, F. X., & Peyré, G. Faster Wasserstein Distance Estimation with the Sinkhorn Divergence. In Neural Information Processing Systems, 2020.
>
> In this paper, we set the maximum iteration number as $\text{Itermax}=200$ in the Sinkhorn algorithm for all experiments, which has been stated in the Experiments section. In the revised version, we have added Fig. 5 in Appendix B.6 to visualize the convergence rate of the Sinkhorn algorithm, where we consider the task about the ``sum of digits''  (DeepSets+OT). The upper figure shows the convergence rate of the Sinkhorn algorithm. The bottom figure visualizes the transport plan matrix for a chosen set. We find that 200 iterations are typically enough for the Sinkhorn algorithm and we can learn a sparse transport plan matrix when the algorithm converges. Notably, the transport plan matrix needs to satisfy two marginal constraints, defined by the probability measures of two distributions, respectively. Recall that the empirical distribution has an unchanged uniform probability measure, so the learned transport plan matrix is dense for the $N_j$ observed samples. In terms of another distribution, its probability measure is the set-specific representation, weighting the importance of K shared centers for the corresponding set. Therefore, it is reasonable that the transport plan matrix is sparse for K centers.
>
> Q4: In the previous experiments, we set $\epsilon$ in Equation (6) as 0.1 for all experiments, which controls the weight of the entropic regularisation in the Sinkhorn algorithm. Notably, unless specified otherwise, we specify the construction of cost function as $C_{ik}=1-\cos \left(\mathbf x_{ji}, \mathbf \beta_{k}\right)$. Therefore, the cost function provides an upper-bounded positive similarity metric, making the $\epsilon$ have the corresponding reasonable range as prior knowledge. Following your suggestion, we further study our DeepSets(+OT)'s sensitivity to $\epsilon$. Here, we consider the point cloud classification task, and each object is represented as a set of $N=20$ vectors. As shown in Fig. 4 of Appendix B.5 in our revised version, we report the performance of DeepSets(+OT) on point cloud classification task with varying $\epsilon$, where DeepSets serves as the baseline. It can be seen that our model is robust to the $\epsilon$. Besides, all the results of DeeepSets(+OT) with different $\epsilon$ are superior to that of DeeepSets, indicating the effectiveness of our method. By fine-tuning $\epsilon$ for each dataset in each task, we might obtain better results than those reported in our experiments. However, we aim to validate our method and thus set $\epsilon=0.1$ instead of exhaustively tuning this hyper-parameter.

---

> > ### Author Response · Authors · 2021-11-22
> > **Response to Reviewer tuTq (3/3)**
> >
> > Summary of the review: The paper seems sound but I have concerns about the novelty: I am willing to raise my score if the authors clarify their contribution w.r.t [1] and [2]. We agree with the ideas of [1-2] are related to ours, all of which assume a trainable reference set, and optimize the distance between original sets (or features of the observed data points) and the reference set with OT theory. However, there are fundamental differences between [1-2] and ours. Connections and comparisons between our model and [1-2] have been discussed in the related work section in the revised version, which are also explained below. The optimal Transport Kernel Embedding (OTKE) [1] marries ideas from optimal transport (OT) theory and kernel methods. Concretely, OTKE embeds the feature vectors of a given set to a reproducing kernel Hilbert space (RKHS) and then performs a weighted pooling operation, with weights given by the transport plan between the set and a trainable reference. Wasserstein Embedding for Graph Learning (WEGL) [2] also uses a similar idea to the linear Wasserstein embedding as a pooling operator for learning from sets of features. Although both [1] and [2] adopt the OT theory and the elements defined in their reference are similar to the shared prototypes (centers) in our work, there are fundamental differences between them and ours in terms of the relations between shared prototypes and set representation. To the best of our knowledge, both of them view the reference distribution as the barycenter and compute the set-specific representation by aggregating the features in a given set with adaptive weight, which is derived from the transport plan between the given set and the reference set. However, we assume $J$ probability distributions (rather than one reference distribution with a uniform measure) over these shared prototypes, where we view $J$ set-specific representations as their measures, to approximate the corresponding $J$ empirical distributions, respectively. Based on our proposed OT distance between empirical distribution and its learnable distribution, we naturally use the existing summary networks as the encoder to compute the set-specific representation. Therefore, the summary network and the global prototypes can be jointly trained by minimizing the OT distance in an unsupervised way. Different from OTKE and WEGL, which adopt kernel-based methods to compute feature of each data point within a set and aggregate them by taking the transport plan as the weight, we naturally use the existing summary networks as the encoder to compute the set's feature. Therefore, we can encode the test set to its representation using the summary network, avoiding iteratively computing the transport plan between the test set and learned reference distribution like OTKE and WEGL. These differences between them and ours lead to different views of set representation learning and different frameworks as well. Moreover, OTKE and WEGL mainly focus on either biological sequence classification tasks or graph classification tasks, while our model provides a promising tool for addressing not only the general set-input problems but also the meta-learning problems.

---

> > > ### Comment · Reviewer_tuTq · 2021-11-22
> > > **Thank you !**
> > >
> > > I appreciate the clarification on the difference between your work and [1], [2]. I raised my score to 6.

---

> ### Author Response · Authors · 2021-11-22
> **Response to Reviewer tuTq (1/3)**
>
> Q1 (Equation (6) seems to be an instance of Wasserstein barycenter) : Thank you very much for pushing us to provide a deeper analysis of our model. We have revisited Eq (6), and we consider Eq (6) as a general OT problem rather than an instance of Wasserstein barycenter. Considering equations and symbols,  we have discussed the differences between our work and the barycenter problem in Appendix F of the revised version.

---

### Official Review · Reviewer_Nrfc · 2021-11-01

**Correctness:** 4
**Technical Novelty And Significance:** 3
**Empirical Novelty And Significance:** 3
**Recommendation:** 8
**Confidence:** 3

**Main Review:**

**Strengths**

The intuition of the work is clear and the approach is reasonable a priori. This makes it easy to follow the motivations and illustrations, and lowers the barrier to the community adopting the presented techniques.

The proposed method is simple and does not require complicated architectural or training adaptations to improve the models to which it is applied.

The experiments are extensive, varied and compelling.

The related work section is mostly good and does a good job of comparing the contained approach with existing work (although see the minor comments.)

**Weaknesses**

The main weakness is in the presentation of the manuscript. I think these problem can be addressed by the authors within the rebuttal period and my score is based on the expectation that this happens; if the authors do not improve the technical aspects of the presentation I will lower my score.

1. (Technical) All the tables should include errors. Currently the tables 1 and 3 do not include errors. The table captions should also include details of the number of repetitions used to calculate the mean and the error that is being reported (standard deviation or standard error on the mean). An indication should be made when results are within error.

2. A meta-statistic would be useful for Table 2. In many cases the original model and the +OT model performances are within error but overall 14 of the 16 head-to-head comparisons results in the OT model having the higher mean. It is unlikely that the addition of the OT loss does not improve performance given such a high number of head-to-head (e.g. in a simple binomial model with $p=0.5$ and 16 trials, the probability of 14 or greater successes is <0.03).

3. (Non-technical) Figure 2 would be improved by using a better colour scheme and different markers for each model to make it easier to distinguish the different models in black and white or for a colour blind person. This is probably an access issue which is why I have included it in the weaknesses rather than the minor comments, and it is in the authors' interest to improve the presentation of their results in any case.

4. (Non-technical) The last sentence of page 2 reads _‘Despite their effectiveness, there is no clear evidence that the output of these summary networks could describe a set’s summary statistics well, which have been proven crucial for the set-structured inference problems (Chen et al., 2021)’_. If X is necessary for Y, and Z has Y, then Z has X. If describing a set’s summary statistics well is crucial for set-structured inference problems, and summary networks are effective at set-structured inference problems, then the output of summary networks describe a set’s summary statistics well. Either effectiveness is clear evidence or describing a set’s summary statistics well is not crucial for these problems. This sentence should be revised.

5. There is no supplementary code provided which means I cannot verify the experimental claims, this has resulted in a lower confidence score in my review.

**Minor comments and suggestions**

The following comments are minor and potentially subjective so feel free to ignore them. My score does not depend on these comments being acted on.
- Infinite Mixture Prototypes for Few-Shot Learning Allen et al ICML 2019 seems like a relevant reference that could be discussed in the related work section on developments to metric based approaches. The approaches differ in fundamental ways, and may be complimentary, but both have the concept of learning multiple centres and it may help to position this work to discuss the ideas presented here in contrast with those presented by Allen et al.
- Algorithm 1 is in the Appendix and that should be stated in the text when it is referenced (e.g. page 4)
- Algorithm 1 should be included in the main text.
- 'as good as possible' on page 5 should be 'as well as possible'
- The related work section has many sentences that should be revised:
  - 'Different from them that focus on...'
  - 'where most related work to ours is...'
  - Different from these studies usually compute class' prototypes...'
- Figure 3 is not particularly compelling and takes up a lot of space that could be better used by presenting Algorithm 1.

**Summary Of The Paper:**

This work introduces a straightforward development for set representation learning in the meta-learning context based on the intuition that the sets encountered in real-world meta-learning tasks tend to have common attributes, as illustrated in Figure 1. The idea is to jointly learn these common attributes $\beta_{[1:K]}$, referred to as 'global centres' or 'global prototypes', and the parameters of a summary network using an Optimal Transport derived loss function that compares the empirical set distribution $P_j$ with a set summary distribution over the global centres/prototypes, $Q_j=\sum_{k=1}^K h_{jk}\delta_{\beta_k}$. The idea is elegant in its simplicity and effectiveness.

**Summary Of The Review:**

In summary, I think this is a well motivated and elegant proposal that has been shown to be effective in a variety of experimental settings. I found the authors' claims to be well supported and I cannot find technical fault with their work that would support rejection, so I will recommend acceptance. The lack of supplementary code reduces my confidence in the review as I can only assess the claims as they are presented. The main issue with the work is its presentation which may be easily improved within the rebuttal.

---

> ### Author Response · Authors · 2021-11-22
> **Response to Reviewer Nrfc**
>
> Thank you for your comments.
>
> Q1: We have added the errors into Tables 1 and 3 in the revised version, where we repeat all experiments $5$ times and report the mean and standard deviation on corresponding test datasets.
>
> Q2: Thanks for your suggestion. For the few-shot classification, we conducted 10000 tasks of the training set to train the model while 1000 tasks of the test set to evaluate the learned models, which was described in Appendix C. That is to say, for the learned model, we calculate its mean accuracy and corresponding standard deviation from 1000 test tasks. To facilitate meta-statistics, we use the two sample T-test to compute the p-value, when comparing our method with its baseline. More specifically, for a pair of samples (our method and its baseline), we use the scipy package in python, which takes the means, standard deviations, and the numbers of observations (1000) of two samples as the input and outputs the p-value. After calculating all p-values for 16 head-to-head comparisons, we find that all p-values are less than $0.05$. We have added the p-value when comparing our model with its corresponding baseline in Table 2 in the revised version, where p-value with blue color indicates an improvement derived from our model and p-value with red color indicates a drop from our model. As we can see, our proposed method enhances its baseline on almost all of the settings, even with slightly lower performance on MatchNet(+OT) (resnet10) and MatchNet(+OT) (resnet34), indicating the effectiveness of our proposed method.
>
> Q3: Following your suggestion, we have re-plotted Figure 2 in the revised version, where we adopt a better color scheme and different markers to improve it.
>
> Q4: We agree with your point. Actually, we had described this sentence more clearly in the Introduction Section, described as
> ``A desideratum of a summary network is to extract set features, which have enough ability to represent the summary statistics of the input set and thus benefit the corresponding set-specific task; but for many existing summary networks, there is no clear evidence or constraint that the outputs of the summary network could describe the set’s summary statistics well. ''. Thus, we have deleted this ambiguous sentence (last sentence of page 2) in the revised version due to the limited space.
> Q5: Following your suggestion, we have uploaded our code as supplementary material.
>
> Minor comments 1: Thanks for this reference. In the revised version, we have discussed this work and compared it with our model in the related work Section on developments to metric-based approaches. We re-elaborate their differences in this response:
> ``Another recent work for learning multiple centers is infinite mixture prototypes (IMP) presented by Allen et al. (2019). The key difference is that IMP represents each class by a set of clusters and infers the number of clusters with Bayesian nonparametrics. In our work, the centers are shared for all classes and set-specific feature extracted from the summary network serves as the proportion of centers, where the centers and summary network can be jointly learned with OT loss.''
>
> Minor comments 2: Following your suggestion, in the revised version, we have moved Figure 3 about the generated natural images to Appendix [E] and moved Algorithm 1 to the main text. Thanks for your careful check. We have corrected these typos in the revised manuscript.

---

### Official Review · Reviewer_TfNJ · 2021-11-02

**Correctness:** 4
**Technical Novelty And Significance:** 2
**Empirical Novelty And Significance:** 3
**Recommendation:** 6
**Confidence:** 3

**Details Of Ethics Concerns:**

‌‌

**Main Review:**

‌‌ **‌Originality**

The paper is original and proposes an Optimal Transport (OT) based algorithm for improving existing summary networks for learning from set-structured data.

**Quality**

The paper is technically sound, however, the novelty is a little bit limited. The proposed approach doesn't improve upon the pre-existing summary networks architecture.

**Clarity**

The paper is clear and easy to follow. Algorithm 1 should be moved to the main body rather than the appendix.

**Significance**

The work is significant, but novelty is a bit limited.

**Limitations**

- The paper lacks theoretical analysis for the proposed approach.
- Error bars are missing and not reported in the results.


**Questions to Authors**

**Summary Of The Paper:**

The paper provides an optimal transport (OT) based algorithm for improving existing summary networks for learning from set-structured data. The proposed approach views each set as a distribution over a set of global prototypes. To learn the distribution over the global prototypes, the proposed approach minimizes the OT distance to the set's empirical distribution over data points. Empirical results demonstrate that the proposed framework improves upon the existing summary network approaches as well as metric-based few-shot classification and generative modeling applications.


**Summary Of The Review:**

‌‌ The paper provides an optimal transport (OT) based algorithm for improving existing summary networks for learning from set-structured data. The paper is original, and  technically sound, however, the novelty is a little bit limited. The proposed approach doesn't improve beyond the pre-existing summary networks architecture. Empirical results demonstrate that the proposed framework improves upon the existing summary network approaches as well as metric-based few-shot classification and generative modeling applications. The paper lacks theoretical analysis for the proposed approach, and the error bars are not reported for the empirical results.

---

> ### Author Response · Authors · 2021-11-22
> **Response to Reviewer TfNJ**
>
> Thanks for your comments. Q1: We agree with you that theoretical analysis is desirable, which however is non-trivial in the context of muti-distributions, summary networks, and others. We are still working on this problem. But, please note that our proposed method is still novel and promising, for providing a new view to addressing set-input and meta-learning problems.  Below, we will clarify our contributions. In this paper, we aim to design a general and flexible method to improve the representation learning from set-structured data, and we have conducted extensive experiments to show the effectiveness of our proposed method.  The basic idea of our approach is very straightforward and intuitive:  different sets in a meta-distribution are closely related and share certain statistical properties. Therefore, we assume there are $K$ shared centers (prototypes). For each set, we further assume a set-specific K-dimensional representation, extracted from a to-be-learned summary network, as the proportion of $K$ shared prototypes. Now, each set-input can be endowed with two distributions: one is the empirical distribution over the data points, and another is the distribution represented with shared prototypes and set-specific K-dimensional representation. Naturally, this work proposes to optimize the summary network (for computing K-dimensional set-specific representation) and $K$ shared prototypes by minimizing OT distance between the two distributions for all sets. Our method is very flexible and can be used to improve existing summary networks by combining the OT loss and the task-specific loss. Thanks to this unsupervised OT loss, the summary networks can also be naturally integrated into other tasks without changing the architecture a lot, such as few-shot classification and few-shot generation, where how to extract representative information from each set (or each class) in an unsupervised way is particularly important. In brief, our method provides a promising tool for addressing set-input and meta-learning problems.
>
>
> Q2: In the revised version, we have added the error bars into Tables 1 and 3, where we repeat all experiments $5$ times and report the mean and standard deviation on corresponding test datasets.

---

### Decision · Program_Chairs · 2022-01-20

**Decision:**

Accept (Poster)

**Comment:**

The proposed method for set representation learning with an application to mete learning is well-motivated and reasonable. Reviewers' original concerns about novelty and technical presentation have been well explained and addressed in the revision. If some theoretical analysis can be provided regarding the proposed method, it would make this work stronger.

In summary, a positive recommendation is given here.